

# Hydromorphological restoration stimulates river ecosystem metabolism

Benjamin Kupilas*[1], Daniel Hering[1], Armin W. Lorenz[1], Christoph Knuth[2], Björn Gücker[3]

[1] Department of Aquatic Ecology, University of Duisburg-Essen, Universitätsstr. 5, D-45141 Essen, Germany

[2] Hydrogeology Department, Ruhr-Universität Bochum, Universitätsstr. 150, D-44801 Bochum, Germany

[3] Department of Geosciences, Applied Limnology Laboratory, Campus Tancredo Neves, Federal University of São João del-Rei, 36301-360 São João del-Rei, MG, Brazil

**\* Author of correspondence**

Email: benjamin.kupilas@uni-due.de

**Keywords**: river restoration, ecosystem function, functional indicators, gross primary production, ecosystem respiration



**Abstract**
Both, ecosystem structure and functioning determine ecosystem status and are important for the
provision of goods and services to society. However, there is a paucity of research that couples
functional measures with assessments of ecosystem structure. In mid-sized and large rivers, effects of
restoration on key ecosystem processes, such as ecosystem metabolism, have rarely been addressed
and remain poorly understood.
We compared three reaches of the third-order, gravel-bed river Ruhr in Germany: two reaches restored
with moderate (R1) and substantial effort (R2) and one upstream degraded reach (D).
Hydromorphology, habitat composition and hydrodynamics were assessed. We estimated gross
primary production (GPP) and ecosystem respiration (ER) using the one-station open-channel diel
dissolved oxygen change method over a 50-day period at the end of each reach. Values for
hydromorphological variables increased with restoration intensity (D < R1 < R2). Restored reaches
had lower current velocity, higher longitudinal dispersion and larger transient storage zones. However,
fractions of median travel time due to transient storage were highest in R1 and lowest in R2, with
intermediate values in D. The share of macrophyte cover of total wetted area was highest in R2 and
lowest in R1, with intermediate values in D. Station R2 had higher average GPP and ER than R1 and
D. The average GPP:ER was significantly higher downstream of restored reaches than of the degraded
reach, indicating increased autotrophic processes following restoration. Temporal patterns of ER
closely mirrored those of GPP, pointing to the importance of autochthonous production for ecosystem
functioning. In conclusion, high reach-scale restoration effort had considerable effects on river
hydrodynamics and ecosystem functioning, which were mainly related to massive stands of
macrophytes. High rates of metabolism and the occurrence of dense macrophyte stands may increase
the assimilation of dissolved nutrients and the sedimentation of particulate nutrients, thereby positively
affecting water quality.



## 1. Introduction

River restoration is a pivotal element of catchment management to counteract anthropogenic

degradation and depletion of river health and water resources, and to increase overall biodiversity and

ecosystem services provisioning (Bernhardt et al., 2005; Strayer and Dudgeon, 2010). Based on

legislative frameworks such as the EU Water Framework Directive (WFD) and the Clean Water Act in

the United States, large investments have been made to restore rivers. In Europe, degraded river

hydromorphology is considered one of the central impacts to the ecological status of rivers (EEA,

2012; Hering et al., 2015). For example, the hydromorphology of about 85% of German rivers is

affected to an extent that they fail to reach the 'good ecological status' demanded by the WFD (EEA,

2012). Accordingly, most restoration projects target the hydromorphological improvement of rivers.

The majority of restoration measures is implemented at the reach-scale, covering short river stretches

typically of one km or less (Bernhardt et al., 2005; Palmer et al., 2014). A variety of reach-scale

measures have been implemented (Lorenz et al., 2012): for instance, restoration activities along

mountainous rivers in central Europe mainly targeted re-braiding and widening of streams, leading to

greater habitat and hydrodynamic heterogeneity (Jähnig et al., 2009; Poppe et al., 2016). In

combination with other characteristics of the river ecosystem – e.g., light, organic matter, nutrient

availability, temperature, hydrologic and disturbance regimes – such hydromorphological changes

likely affect biological community composition and ecosystem functioning, including ecosystem

metabolism (Bernot et al., 2010; Tank et al., 2010).

The assessment of restoration effects has mainly focused on responses of aquatic organisms, such as

fish (e.g., Roni et al., 2008; Haase et al., 2013; Schmutz et al., 2016), benthic invertebrates (e.g.,

Jähnig et al., 2010; Friberg et al., 2014; Verdonschot et al., 2016), and macrophytes (e.g., Lorenz et al.,

2012; Ecke et al., 2016). Recently, increasing attention has also been given to the response of

floodplain organisms (e.g., Hering et al., 2015; Göthe et al., 2016; Januschke and Verdonschot, 2016),

while functional characteristics, i.e. the rates and patterns of ecosystem processes, have rarely been

addressed. Ecosystem functions are life-supporting processes that are directly linked to ecosystem

services, i.e. the benefits people obtain from the environment (Palmer and Filoso, 2009). Thus, an





emerging interest in river restoration research is to incorporate the recovery of ecological functioning
(Palmer et al., 2014). However, only few studies have considered the response of river ecosystem
functioning and functional metrics to restoration (e.g., Lepori et al., 2005; Bunn et al., 2010; Kupilas et
al., 2016). Consequently, the effects of restoration on key ecosystem processes remain poorly
understood.
Ecosystem metabolism, i.e. the combination of gross primary production (GPP) and ecosystem
respiration (ER), is a fundamental ecosystem process in rivers. It measures the production and use of
organic matter within a river reach by all biota. Therefore, it provides key information about a river's
trophic and energetic base (relative contribution of allochthonous and autochthonous carbon) (Young
et al., 2008; Tank et al., 2010; Beaulieu et al., 2013). The majority of stream ecosystem metabolism
work investigated natural changes, such as effects of floods and droughts (e.g., Uehlinger, 2000),
seasonal or inter-annual changes (e.g., Uehlinger, 2006; Beaulieu et al., 2013), interbiome differences
(e.g., Mulholland et al., 2001), or land-use change (e.g., Gücker et al., 2009; Silva-Junior et al., 2014).
The majority of these studies focused on smaller streams, while only few studies measured metabolism
of larger streams and rivers (e.g., Uehlinger, 2006; Hall et al., 2016). The response of stream
metabolism to hydromorphological changes, e.g. through river widening, is almost unknown,
especially for larger rivers (but see Colangelo, 2007).
The widening of the riverbed enhances habitat complexity and diversity of the river channel and the
riparian zone (Jähnig et al., 2010; Januschke et al., 2014; Poppe et al., 2016). Moreover, channel
widening also favors macrophytes and other autotrophs through the creation of shallow, slow flowing
areas and backwaters (Lorenz et al., 2012). Further, it increases light availability and water
temperature, which have been identified as major factors controlling river metabolism, especially
primary production (Uehlinger, 2006; Bernot et al., 2010; Tank et al., 2010). Accordingly, these
changes potentially lead to enhanced in-stream autotrophic processes.
Restoration also increases the retention of allochthonous organic matter (Lepori et al., 2005; Lepori et
al., 2006; Flores et al., 2011). Moreover, the reconnection of rivers with their floodplains by creating
shallower river profiles and removing bank fixations may enhance inundation frequency, and hence



resource transfers from land to water. In combination, these changes can favor heterotrophic activity in
the river. Restoration also affects hydrodynamics and surface water-ground water interactions of
streams (Becker et al., 2013): for instance, widening of the stream channel reduces flow velocity and
the creation of backwaters and pools possibly leads to changes in the size and location of transient
storage zones (Becker et al., 2013). Though previous studies revealed an inconsistent relationship
between hydrodynamics and metabolism (Beaulieu et al., 2013), increases in transient storage zones
potentially enhance ER (Fellows et al., 2001) and nutrient processing (Valett et al., 1996; Gücker and
Boëchat, 2004).
The objective of this study was to quantify reach-scale restoration effects on hydromorphology, habitat
composition and hydrodynamics, as factors potentially affecting river ecosystem function, by
comparing three continuous stream reaches (two restored and one upstream non-restored reach) of a
mid-sized mountain river in Germany and to determine the corresponding responses of river
metabolism. We expected (i) hydromorphological river characteristics, i.e., habitat composition and
hydrodynamics to change concomitantly with restoration (e.g. wider and more diverse river channel,
and higher abundance of primary producers in restored river reaches compared to the degraded reach,
as well as changes in the sizes and locations of transient storage zones). Further, we expected (ii)
ecosystem metabolism to respond with increased metabolic rates, i.e. enhanced GPP and ER, mainly
as a result of increased abundances of primary producers.
**2. Methods**
*2.1 Study site*
This study was conducted in the upper River Ruhr (Federal State of North Rhine-Westphalia,
Germany, Fig. 1, Table 1) a tributary to the Rhine. The third-order Ruhr is a mid-sized mountain river
with gravel and cobbles as bed sediments. The catchment area upstream of the study site is 1060 km²,
about 64 % of which is forested, 28 % is arable land and pasture, and 8 % is urban area (located
mainly in the floodplains). The study site is at an altitude of 153 m a.s.l. and the mean annual
discharge was 21.3 m³ s⁻¹ between 2004 and 2009. The Ruhr is draining one of the most densely





populated areas of Europe; however, population density of the upstream catchment area is low (135.3
inhabitants/km² upstream of the study site). Due to manifold uses, the river's hydromorphology has
been largely modified by impoundments, residual flow sections, bank fixation as well as industrial and
residential areas in the floodplain. More recently, the hydromorphology of several river sections has
been restored.
Restoration aimed to establish near-natural hydromorphology and biota. Restoration measures
included the widening of the riverbed and the reconnection of the river with its floodplain by creating
a shallower river profile and by removing bank fixations. Moreover, the physical stream quality was
enhanced by generating secondary channels and islands, adding instream structures, such as woody
debris, and creating shallow habitats providing more space for autotrophs (see Appendix S1 in
Supporting Information).
We separated the restored reach into two reaches of approximately similar lengths (1210 and 1120 m)
with obvious differences in morphological stream characteristics due to differing restoration effort
(R1: moderate restoration effort and R2: high restoration effort). Briefly, in R2 a larger amount of soil
was removed and the costs for the implementation of measures were higher than in R1 (see Appendix
S1). In R2 the bank fixation was removed at both shorelines and the river was substantially widened
and secondary channels and islands were created, while the removal of bank fixation and widening in
R1 mainly focused on one site due to constrains posed by a nearby railroad (see Appendix S1). The
restored reaches were compared to a degraded "control-section" of 850 m length located upstream of
the restored reaches (D). The degraded reach was characteristic for the channelized state of the River
Ruhr upstream of the restoration site, and reflected the conditions of the restored sections prior to
restoration: The reach was a monotonous, channelized and narrowed river section with fixed banks
and no instream structures. A 650 m-long river section separating the degraded from the restored river
reach was excluded from the investigations, as its hydromorphology was deviating due to
constructions for canoeing and a bridge. As the three sections were neighboring each other, differences
in altitude, slope, discharge and catchment land cover between reaches were negligible.
*2.2 Hydromorphology and habitat composition*



Physical stream quality was quantified from aerial photos. High-resolution photos of the restored
reaches were taken in summer 2013 using a Falcon 8 drone (AscTec, Germany). Aerial photos of the
degraded reach from the same year at similar discharge conditions were provided by the Ministry for
Climate Protection, Environment, Agriculture, Conservation and Consumer Protection of the State of
North Rhine-Westphalia. Photos were analyzed in a geographical information system (ArcGIS 10.2,
ESRI). For each reach, we measured the width of the wetted channel every 20 m along cross-sectional
transects and calculated mean width and its variation (reach D: n = 42, R1: n = 59, R2: n = 54). For
each reach, we recorded thalweg lengths, the area of the wetted stream channel, the floodplain area
(defined as bank-full cross-sectional area), and the area covered by islands, woody debris, and aquatic
macrophyte stands (Fig. 2). Subsequently, the share of macrophyte stands of the total wetted area was
calculated for each reach. Additionally, macrophytes were surveyed according to the German standard
method (Schaumburg et al., 2005a; b) in summer 2013. A 100 m reach was investigated by wading
through the river in transects every 10 m, and walking along the riverbank (Lorenz et al., 2012). All
macrophyte species were recorded and species abundance was estimated following a 5-point scale
developed by Kohler (1978), ranging from "1 = very rare" to "5 = abundant, predominant". The
empirical relationship between the values of the 5-point Kohler scale (x) and the actual surface cover
of macrophytes (y) is given by the function $y = x^3$ (Kohler and Janauer, 1997; Schaumburg et al.,
2004). Using this relationship, we $x^3$-transformed the values of the Kohler scale into quantitative
estimates of macrophyte cover for the studied 100 m reaches.
*2.3 Hydrodynamics*
Stream hydrodynamics were estimated using a conservative tracer addition experiment with the
fluorescent dye Amidorhodamine G. Across the river width, we injected the dissolved dye in a
distance sufficiently upstream to the first study reach to guarantee complete lateral mixing at the first
sampling station. Breakthrough curves of the tracer were continuously measured in the main current at
the upstream and downstream ends of all three reaches (Fig. 1). Concentration of dye was recorded at
a resolution of 10 s at the most upstream and downstream sampling stations using field fluorometers
(GGUN-FL24 and GGUN-FL30, Albillia, Switzerland). At the other sampling stations (start and end



of each investigated river reach) water samples were taken manually at 2 min intervals. The samples
were stored dark and cold in the field and subsequently transported to the hydrogeochemical
laboratory of the Ruhr University Bochum. Amidorhodamine G concentrations of water samples were
measured with a fluorescence spectrometer (Perkin Elmer LS 45; detection limit of 0.1 ppb) and
standard calibration curves prepared from the tracer and river water. Field fluorometers were
calibrated prior to experiments with the same standard calibration procedure.
Subsequently, we used the one-dimensional solute transport model OTIS-P (Runkel, 1998) to estimate
parameters of river hydrodynamics for each reach from the breakthrough curves: advective velocity,
longitudinal dispersion, stream channel and storage zone cross-sectional areas, and storage rate. We
further calculated fractions of median travel time due to transient storage ($F_{med}^{200}$) based on the
hydrodynamic variables obtained from transport modeling (Runkel, 2002). Additionally, Damköhler
numbers were estimated for each reach (Harvey and Wagner, 2000).
*2.4 Discharge*
Discharge data were provided by the North Rhine-Westphalia State Agency for Nature, Environment
and Consumer Production, Germany (Landesamt für Natur, Umwelt und Verbraucherschutz
Nordrhein-Westfalen) for a gauging station situated at the downstream end of the study site. At this
station, discharge was constantly recorded at 5-min intervals.
*2.5 Ecosystem metabolism*
We estimated river dissolved $O_2$ (DO) metabolism using the "open-channel one-station diel DO
change technique" (Odum, 1956; Roberts et al., 2007). We chose this method instead of the two-
station technique (Marzolf et al., 1994; Young and Huryn, 1998), as the studied reaches were too short
to reliably estimate ecosystem metabolism with the latter method due to high current velocities and
low reaeration rates. Reach lengths influencing the one-station diel dissolved $O_2$ change technique in
our study were typically much longer than the experimental reaches, due to high current velocities and
low reaeration (>10 km; estimated according to Chapra and Di Torro, 1991). Following methods in
Demars et al. (2015), metabolism estimates at the downstream sampling station R2 were only to 35%



influenced by the restored river sections, but to 65% by upstream degraded river sections.
Accordingly, differences in metabolic rates among sampling stations at the end of restored and
impacted experimental reaches as estimated in our study are likely to be much lower than actual
differences among the shorter experimental reaches, and should thus be viewed as qualitative
indicators of restoration effects, rather than measured metabolic rates of the experimental reaches. The
selected method is based on the assumption that changes in DO within a parcel of water traveling
downstream can be attributed to metabolism (photosynthesis and respiration) and to gas exchange
between water and atmosphere, given that no significant groundwater dilution of river water occurs
along the studied river. The change in DO was estimated as the difference between consecutive 5-min
readings at one station (Roberts et al., 2007; Beaulieu et al., 2013).
In two consecutive field campaigns in summer 2014, DO and water temperature were continuously
measured at the downstream ends of the three reaches at 5-min intervals for 50 days. The DO probes
with data loggers ($O_2$-Log3050-Int data logger Driesen + Kern GmbH, Germany) were installed in the
thalweg of the river in the middle of the water column. The DO probes were calibrated in water-
saturated air prior to measurements. Additionally, probes were cross-calibrated for one hour at a single
sampling station in the river before and after the measurements. We used the data of this comparison
to correct for residual differences among probes (Gücker et al., 2009). This procedure assured that
differences between probes were only due to differences in DO and water temperatures and not to
analytical errors. In previous laboratory tests, the probes showed no drift and were thus not corrected
for drift during the measurement campaigns (Almeida et al., 2014).
In parallel to DO and water temperature, atmospheric pressure was recorded (Hobo U20-001-04;
Onset Computer Corporation). We used atmospheric pressure and water temperature data to calculate
the oxygen saturation. Reaeration coefficients ($K_{oxy}^{20}$; standardized for 20°C) were estimated using the
nighttime regression approach (Young and Huryn, 1999). For the downstream stations of all three
sampling reaches, we calculated regressions between DO change rates and DO deficits at night (night
hours were defined as the period 1 h after sunset to 1 h before sunrise). We only considered significant
nighttime regressions ($p < 0.05$). Reaeration coefficients for days without significant regressions were



estimated as the average value of the coefficients of the days before and after, as we did not observe
$K_{oxy}{}^{20}$ - discharge relationships in our data (see Appendix S2) that could have been used to estimate
$K_{oxy}{}^{20}$ values for days without reliable estimates. Estimated reaeration coefficients were low and
ranged from 5 to 15 $d^{-1}$ in our study (see Appendix S2). Subsequently, we calculated ecosystem
respiration (ER) and gross primary production (GPP) as detailed in Roberts, Mulholland & Hill (2007)
from the recorded nighttime river water DO deficit and the daytime DO production, respectively,
corrected for atmospheric reaeration (see Appendix S3). Metabolic rates obtained by this method
closely matched those obtained with the estimator of Reichert et al. (2009). Ground water dilution was
not detected, i.e. discharge differences among the investigated river reaches were within the ranges of
method uncertainty of discharge measurements, and was thus not considered into our estimates.
Metabolism measurements from days at which floating macrophytes accumulated around probes and
affected DO measurements were eliminated from the dataset.
*2.6 Data analysis*
We used repeated measures ANOVAs and Tukey's HSD post-hoc tests to test for differences in
metabolic rates (GPP, ER, NEP, GPP:ER) among sampling stations, comparing daily metabolic rates
among reaches. Data recorded at the time of flooding events were omitted from analyses, because
metabolic rates were not representative (e.g. no detectable GPP); overall, data of n = 32 days were
used in the analyses. Repeated measures ANOVAs and Tukey's HSD post-hoc tests were also used to
test for differences in water temperature among river reaches. Conventional one-way ANOVA was
used to test for differences in river width, comparing the transect measurements performed in the three
river reaches. All statistical analyses were conducted in R (R Development Core Team, 2007).
**3. Results**
*3.1 Hydromorphology and habitat composition*
Restored river reaches were morphologically more complex and had significantly wider wetted
channels (ANOVA and Tukey post-hoc test, P < 0.05) and more variable channel width than the
degraded reach (Table 2). Furthermore, the restored reaches had larger wetted channel areas,



floodplain areas, island areas and patches of woody debris than the degraded river reach (Table 2). The
intensively restored reach R2 showed the highest values for hydromorphological variables (Table 2).
The share of macrophyte cover of total wetted area was also highest in R2.
*3.2 Hydrodynamics*
The reaches differed in hydrodynamic parameters: The restored reaches had lower flow velocity and
higher longitudinal dispersion, cross-sectional areas of the advective channel, and storage zone cross-
sectional areas than the degraded reach (Table 2). Storage rate and fractions of median travel time due
to transient storage ($F_{med}^{200}$) was highest in R1 and lowest in R2, with intermediate values for D (Table
2). Damköhler numbers between 0.5 and 5.0 indicated reliable transient storage parameter estimates
for the reaches (Harvey and Wagner, 2000; Table 2). Tracer breakthrough curves estimated by
transport modelling closely corresponded to measured tracer concentrations (Fig. 3).
*3.3 Discharge and water temperature*
Mean discharge during the first weeks of measurement was 8.4 m$^3$ s$^{-1}$. The hydrograph was
characterized by a large summer flow peak and two minor peaks during the study period (Fig. 4 a).
During the flow peaks discharge rapidly increased 3.5- to 7-fold, relative to the mean flow. Trends in
water temperature over time were very similar for the three river reaches and are exemplarily shown
for R2 (Fig. 4 b). Overall, restored reaches had higher mean daily water temperatures than the
degraded reach, with R2 having higher mean daily water temperatures compared to R1 (repeated
measures ANOVA, $P < 0.0001$; and Tukey's HSD post-hoc tests, $P < 0.0005$).
*3.4 Ecosystem metabolism*
We observed significant effects of reach-scale restoration on metabolic rates estimated at the
downstream ends of restored and degraded reaches. The three sampling stations at the downstream
ends of the reaches generally exhibited similar metabolism patterns (Fig. 5). Rates of GPP and ER
ranged from 2.59 to 13.06 and -4.96 to -17.52 g O$_2$ m$^{-2}$ day$^{-1}$ at sampling station D, from 2.33 to 12.36
and -4.04 to -14.02 g O$_2$ m$^{-2}$ day$^{-1}$ at station R1, and from 3.61 to 17.64 and -5.91 to -24.71 g O$_2$ m$^{-2}$
day$^{-1}$ at station R2. Daily rates of GPP were highest shortly before the main summer flow peak at all



sampling stations (Fig. 5 a). GPP was not detectable during the summer flow peaks. ER generally
mirrored the GPP patterns, but showed distinct peaks at the beginning of the summer flow peak. ER
exceeded GPP during all but one day at R1 and two days at R2. Consequently, NEP (net ecosystem
production) was negative during most of the measured period, i.e. reaches were heterotrophic (Fig. 5
b). The mean GPP:ER ratio ranged from 0.66 to 0.80 across all sampling stations, also indicating that
the Ruhr was moderately heterotrophic. General patterns in daily rates of both GPP and ER also
seemed to be influenced by flow peaks. GPP and ER were both suppressed immediately following the
flooding events. The ensuing recovery patterns for GPP and ER were similar for all investigated
sampling stations: depending on magnitude of flow, GPP and ER were suppressed for several days,
but steadily returning to pre-disturbance conditions.
According to repeated measures ANOVAs of all metabolism estimates excluding those during the
flood events (P < 0.01; and Tukey's HSD post-hoc tests, P < 0.005), sampling station R2 showed
significantly higher GPP and ER than the other stations (Fig. 6). The GPP:ER ratio was significantly
higher at stations R1 and R2 than at station D. NEP was higher at sampling station R1 than at D.
**4. Discussion**
Restoration of river hydromorphology usually covers short river stretches of less than one km and is
expected to increase the river's habitat and hydrodynamic heterogeneity. Together, these changes may
stimulate ecosystem metabolism, i.e. whole-stream rates of GPP and ER, as well as affect the river's
metabolic balance. Increases in river metabolism, in turn, may result in increased rates of other
ecosystem processes, such as secondary productivity and whole-stream nutrient processing (Fellows et
al., 2006; Gücker and Pusch, 2006).
*4.1 Hydromorphological characteristics*
Recent monitoring and evaluation of restoration projects report positive effects on hydromorphology
and habitat composition (Jähnig et al., 2009; Jähnig et al., 2010; Poppe et al., 2016). Similarly, we
found greater habitat complexity of restored reaches, as indicated by wider and more diverse river
channels. The reach with the highest restoration effort (R2), was characterized by the highest values



and heterogeneity of hydromorphological variables; this suggests that restoration effort is indeed
crucial for restoration success. According to Lorenz et al. (2012), the success of restoration in mid-
sized to larger rivers can also be indicated by increased cover, abundance and diversity of macrophytes
as they benefit from more natural and diverse substrate, and the variability in flow. Consequently, the
higher share of macrophyte cover of total wetted area in R2 also highlighted the higher morphological
quality of this reach.
Changes in hydromorphology and habitat composition influenced hydrodynamics: we observed lower
current velocity, higher longitudinal dispersion and larger transient storage zones in the restored
reaches. This corresponds with the larger river width and wetted channel area, and the increased
abundance of morphological features such as woody debris, islands and macrophyte patches.
However, $F_{med}^{200}$, i.e. the relative importance of transient storage for whole-stream hydrodynamics,
was highest in R1 and lowest in R2, with intermediate values for D. Accordingly, there appeared to be
an inverse relationship between $F_{med}^{200}$ and the share of macrophyte cover of total wetted area, which
was highest in R2 and lowest in R1, with intermediate values in D. These findings suggest that the
dense stands of macrophytes in R2 particularly altered stream hydrodynamics: macrophyte patches
built large surface transient storage areas and potentially changed the locations of transient storage
zones from the hyporheic zone to the surface water column. Macrophyte fields in R2 may have even
been so dense that large parts of them were representing hydrodynamic dead zones. A similar effect
was found in streams restored by implementing steering structures to enhance stream quality: the
restored reaches were dominated by surface transient storage exchange (Becker et al., 2013).
Furthermore, the sedimentation of fine sediment within dense macrophyte stands may further decrease
exchange with the hyporheic zone.
*4.2 Functional characteristics*
Metabolism was measured over a 50-day period to obtain representative data, allowing for
comparisons among sampling stations. Furthermore, this time series allowed for the analysis of
environmental variability, such as flow peaks. The results were obtained for the summer period, i.e.
the time of maximum biomass, which is also relevant for the WFD compliant sampling period (e.g.,



Haase et al., 2004; Schaumburg et al., 2004; EFI+ CONSORTIUM, 2009). Therefore, results obtained
in this study are directly comparable to the river status derived from biological assessment.
In general, the three sampling stations showed similar patterns in metabolism, as our one-station
metabolism approach measured a long upstream river section in addition to the experimental reaches.
Rates of ER mirrored those of GPP, suggesting that autotrophic respiration largely drove temporal
patterns in ER, despite an overall ratio of GPP:ER < 1 and a slightly negative NEP during most of the
measurement period. Similar patterns were found in streams in the US (Beaulieu et al., 2013; Hall et
al., 2016). The average GPP:ER ratio was significantly higher downstream of the restored reaches in
our study (0.77 and 0.80, respectively) than downstream of the degraded reach (0.66), indicating an
increase in autotrophic processes following restoration. The only moderate heterotrophic state of the
river together with ER closely tracking GPP indicated the importance of autochthonous production for
the metabolism. This is further supported by the comparison of pre- and post-peak flow ER (Fig. 5).
McTammany et al. (2003) suggested that higher inputs of allochthonous material may occur after
flooding events, subsequently supporting high rates of ER. In line with this, we expected high rates of
ER during the last third of the sampling period, especially in restored reaches with a potentially high
POM trapping efficiency. However, ER was lower compared to pre-flow peak conditions, with ER
still mirroring GPP, thus indicating the coupling of autochthonous production with ER even after
floods. This implies that restoration (reconnection of river and floodplain) did not increase resource
transfer into the channel to such an extent that it influenced river metabolism.
We observed significantly higher GPP and ER at station R2 compared to the other stations.
Metabolism of R1 did not markedly differ from D, corresponding with consistently higher values of
hydromorphological variables in R2 only. Given the previously discussed importance of
autochthonous production for the metabolism, habitat enhancement supporting the growth of
macrophytes is likely the cause for higher GPP and ER in R2. Consequently, only high restoration
effort bringing a restored reach close to reference conditions led to pronounced effects on ecosystem
metabolism. Restoration effects were mainly related to the growth of aquatic macrophytes, which
formed dense stands that augmented ecosystem metabolism. We acknowledge that metabolism was



measured during summer, i.e. the time of maximum biomass of aquatic macrophytes. Therefore, high
GPP and ER measured in this campaign might be restricted to this season and effects will be lower
during winter times when macrophyte abundance will be low.
Ecosystem metabolism of the sampling stations at the restored reaches was expected to be at similar
levels to those of natural rivers reported in the literature. Therefore, we compared GPP and ER of our
sampling stations to those of rivers comparable in size (discharge between 5 - 50 m³ s$^{-1}$; see Appendix
S4, S5). GPP and ER estimated in this study were among the highest values reported for similar sized
rivers; especially those of the sampling station R2. However, there is a tremendous variability in
ecosystem metabolism among natural river reaches in the literature (see Appendix S4, S5).
Considering the limited knowledge about natural geographical gradients in river metabolism, it was
not possible to assess if values obtained for restored reaches indicate natural conditions in a broader
geographic context. In future analyses of restoration effects on fluvial metabolism, local reference
conditions should therefore be assessed whenever possible.
Our experimental reaches reflected typical spatial scales on which restoration measures are
implemented. However, these reaches were too short to feasibly use the two-station diel DO change
method (see 2.5). Accordingly, we used the one-station approach to assess reach-scale restoration
effects on ecosystem metabolism of longer river sections (>10 km). Following methods in Demars et
al. (2015), we evaluated to which extent our metabolism estimates reflected the restored river sections.
Measurements at sampling station R1 and R2 were only to 16% and 24%, respectively, influenced by
the restored experimental reaches directly upstream. However, station R2 was to 35% influenced by
the combined reaches R1+R2, and thus to 65% by upstream degraded river sections. Despite this
mismatch between lengths of river reaches evaluated and reaches exclusively affected by restoration,
we found significant effects of reach-scale restoration on whole-river metabolism. Interestingly, our
study therefore also shows that high restoration effort in short river reaches (1 to 2 km) had
considerable effects on total whole-river metabolic rates of river stretches exceeding the length of the
actually restored reaches (>10 km). Thus, the restoration of short river reaches to near-natural
conditions may have positive effects on downstream river sections regarding diel DO variability and



carbon spiraling. High rates of metabolism and the occurrence of dense macrophyte stands in restored
river reaches may also increase the assimilation of dissolved nutrients (Fellows et al., 2006; Gücker et
al., 2006) and the sedimentation of particulate nutrients (Schulz and Gücker, 2005), thereby positively
affecting water quality.
*4.3 Recommendations for restoration monitoring*
For most regions and river types, data is missing indicate metabolic rates of good, moderate or poor
river conditions. However, based on data from mainly small streams, Young et al. (2008) proposed a
useful framework to assess functional stream health using GPP, ER, NEP and GPP:ER. Consequently,
metabolic rates for different river types should be surveyed to allow the incorporation of ecosystem
metabolism of mid-sized and large rivers as functional indicator in this framework. Our study stresses
the benefits of metabolism as a functional indicator complementing the monitoring of restoration
projects (compare Young et al., 2008; Bunn et al., 2010): Temporally high-resolution and automated
monitoring, that integrates biotic and abiotic variables over time and across habitats may increase our
understanding of the effects of river restoration and might help identifying initial changes after
restoration. Incorporating functional indicators into monitoring programs may enable a more holistic
assessment of river ecosystems and elucidate responses to restoration (and also impairment), which
may be related to ecosystem structure and function.
**Acknowledgements**
We thank D. Dangel, K. Gees, M. Gies, A. Gieswein, K. Kakouei, B. Rieth, L. Rothe, C. Sondermann,
M. Sondermann, and colleagues from the Ruhr University Bochum for their help during the tracer
experiment. We thank L. Wenning and J. Herold for their assistance in many of our field trips, R.
Dietz and D. Hammerschmidt for supporting our research, and the NZO GmbH for providing aerial
photos. We thank B. Demars for helpful comments on a previous version of this manuscript. We also
gratefully acknowledge a PhD fellowship of the German Environment Foundation (Deutsche
Bundesstiftung Umwelt, DBU) to B.K. and a productivity grant by the Brazilian National Council for
Scientific and Technological Development (CNPq 302280/2015-4) to B.G.. This study was financially



supported by the EU-funded project REFORM (Restoring rivers FOR effective catchment
Management), European Union's Seventh Programme for research, technological development and
demonstration under Grant Agreement No. 282656.



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



**Table 1:** River and study site characteristics

| River characteristics | |
|---|---|
| Catchment size (km²) | 4485 |
| Stream length (km) | 219 |
| River type | Gravel-bed |
| Stream order | 3 |
| Ecoregion | Central Highlands |
| **Study site characteristics** | |
| Latitude (N) * | 51.44093 |
| Longitude (E ) * | 7.96223 |
| Catchment size (km²) | 1060 |
| Altitude (m a.s.l.) | 153 |
| Mean annual discharge (m³ s$^{-1}$) | 21.3 |
| Catchment geology | siliceous |
| Restoration length (km) | 2.3 |
| Restoration date | 2007-2009 |
| Main restoration action | riverbed widening |
| pH ** | 8.3 |
| Electric conductance ** ($\mu$ S cm$^{-1}$) | 340 |
| Total nitrogen ** (mg L$^{-1}$) | 2.7 |
| NO$_3$-N ** (mg L$^{-1}$) | 2.53 |
| NH$_4$-N ** (mg L$^{-1}$) | < 0.1 |
| Total phosphorus ** (mg L$^{-1}$) | 0.07 |
| Total organic carbon ** (mg L$^{-1}$) | 2.3 |

* center of reach
** data from ELWAS-WEB (online information system maintained by The Ministry for Climate Protection, Environment, Agriculture,
Conservation and Consumer Protection of the State of North Rhine-Westphalia; sampling date: 26.6.2012).



**Table 2**: Morphological and hydrodynamic characteristics of the investigated river reaches

| Variable | degraded reach (D) | 1. restored reach (R1) | 2. restored reach (R2) |
|---|---|---|---|
| Thalweg length (m) | 850 | 1210 | 1120 |
| Width (m) | 22.5 | 28.2 | 36.6 |
| Width variation * (m) | 3.3 | 6.3 | 10.5 |
| Wetted channel area (m²) | 19,114 | 34,604 | 41,673 |
| Floodplain area (m²) | 27,363 | 30,630 | 34,218 |
| Island area (m²) | 0 | 2,666 | 12,381 |
| Woody debris (m²) | 0 | 467 | 691 |
| Macrophyte coverage (%) | 4.8 | 1.7 | 19.8 |
| Flow velocity (m s$^{-1}$) | 0.95 | 0.8 | 0.47 |
| Longitudinal dispersion, $D$ (m² s$^{-1}$) ** | 0.28 | 0.59 | 10.21 |
| Channel cross-sectional area, $A$ (m²) ** | 12.11 | 14.96 | 27.05 |
| Storage zone cross-sectional area, $A_S$ (m²) ** | 2.38 | 4.48 | 3.16 |
| Storage rate, $\alpha$ (s$^{-1}$) ** | $4.9 \times 10^{-4}$ | $7.4 \times 10^{-4}$ | $2.0 \times 10^{-4}$ |
| Transient storage, $F_{med}^{200}$ (%) | 1.6 | 3.9 | 0.8 |
| Damköhler number | 2.8 | 4.8 | 4.4 |

* Width variation calculated as standard deviation; degraded: n = 42, restored 1: n = 59, restored 2: n = 54
** Data on hydrodynamic characteristics represent the final parameters obtained by one-dimensional transport modelling using OTIS-P.





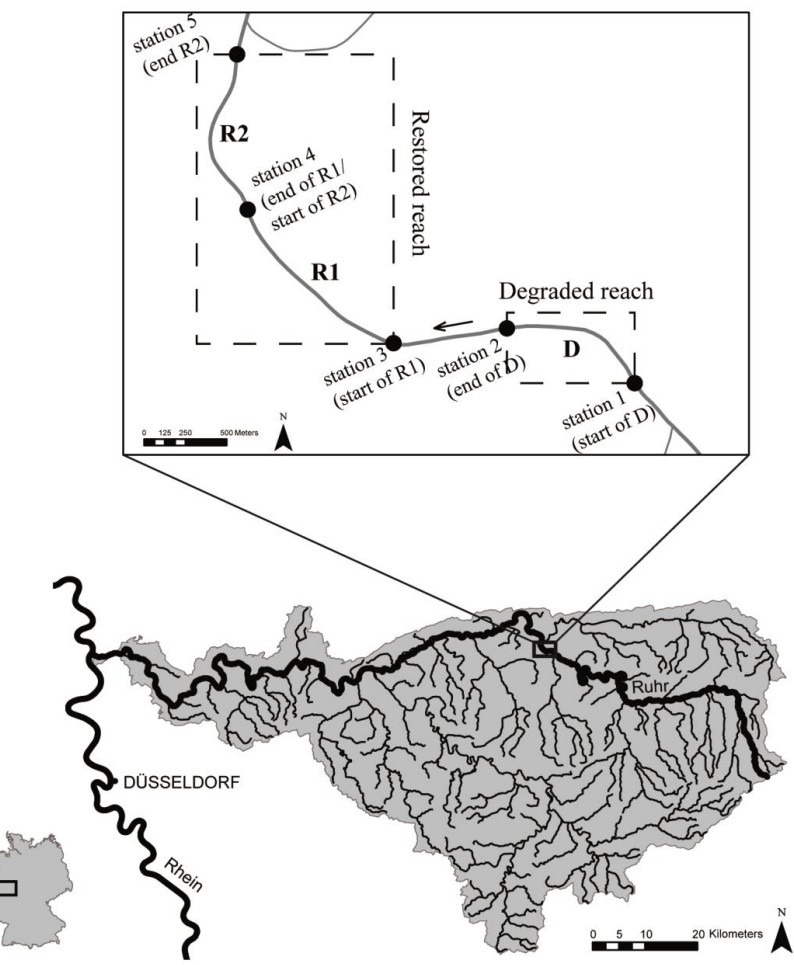


**Fig. 1:** Location of the study site in the upper catchment of the River Ruhr in Germany. Stations represent start and end of the

investigated river reaches (degraded, 1st restored and 2nd restored reach).





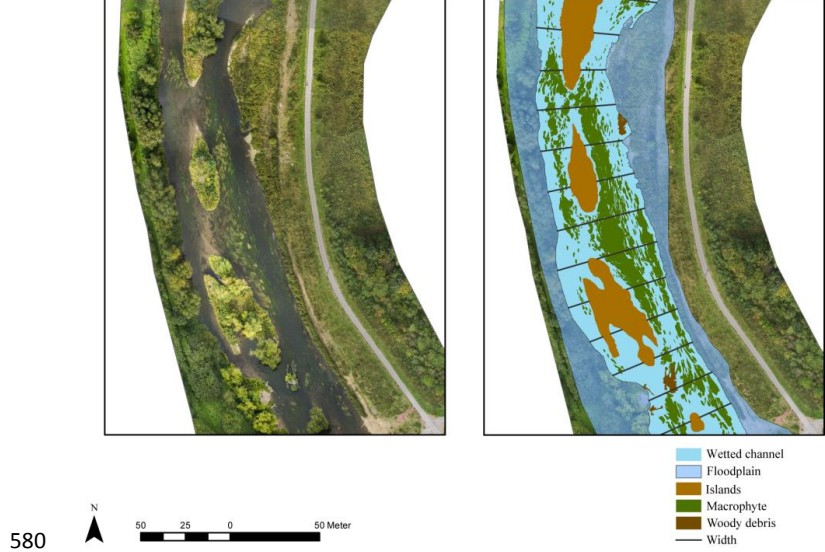


**Fig. 2:** Analysis of aerial photos. A representative river section of the 2$^{nd}$ restored reach is shown.

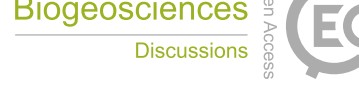



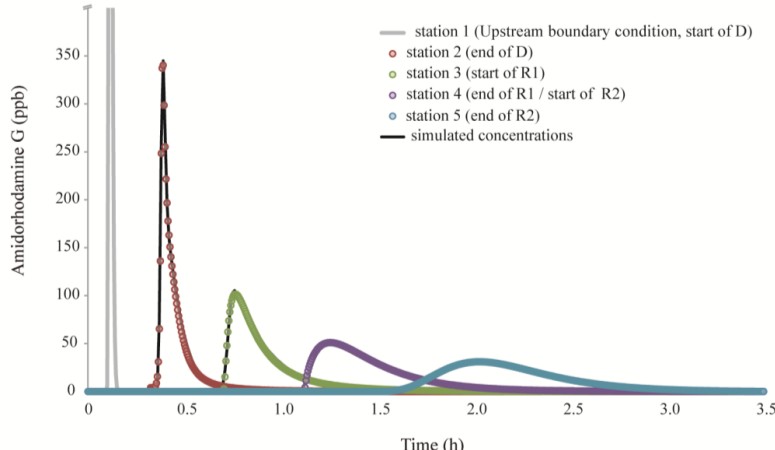


**Fig. 3:** Tracer breakthrough curves for the conservative tracer addition experiment in the River Ruhr. Upstream boundary
condition based on concentrations at sampling station 1 (start of degraded reach, D, grey solid line), observed concentrations
at sampling stations 2 (end of degraded reach, empty circles), 3 (start of 1[st] restored reach, R1, empty squares), 4 (end of 1[st]
restored reach, start of 2[nd] restored reach, R2, empty triangles), 5 (end of 2[nd] restored reach, crosses), and simulated
concentrations based on final parameter estimates with OTIS-P (solid lines).


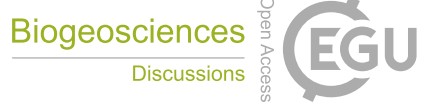

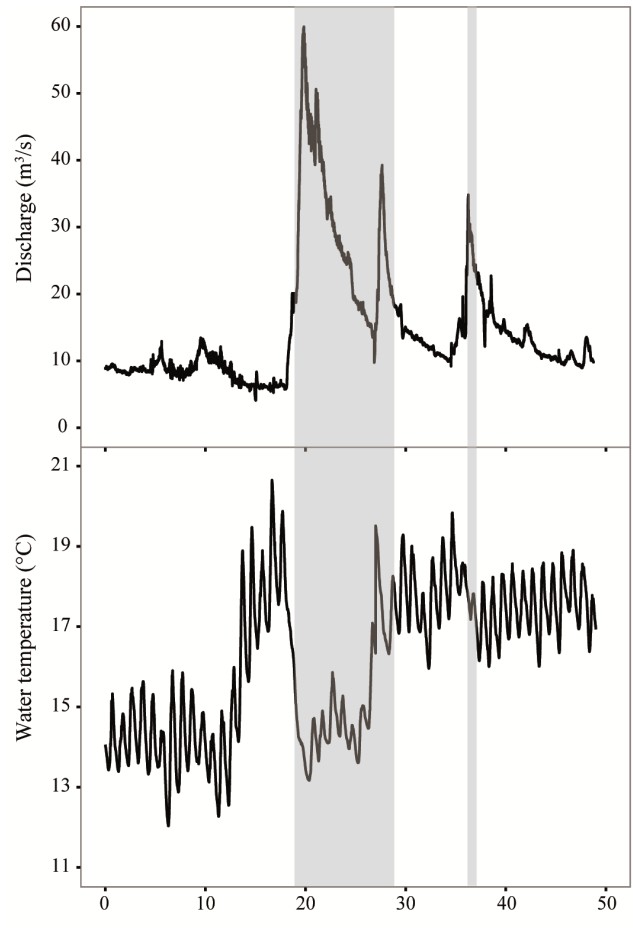

588

**Fig. 4:** (a) discharge and (b) water temperature in the River Ruhr during the study period in summer 2014. Trend in water
temperature during study period is exemplarily shown for the 2nd restored reach (R2).





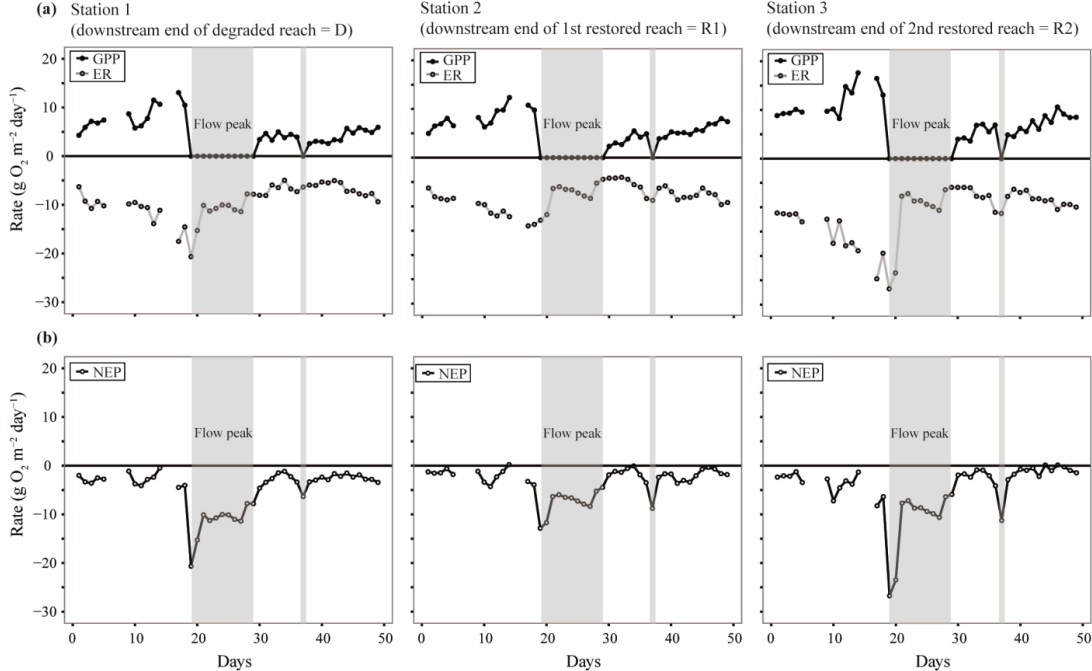

591

**Fig. 5:** Daily rates of (a) gross primary production (GPP: positive values, black line) and ecosystem respiration (ER: negative

values, grey lines) and (b) net ecosystem production (NEP) measured at the downstream ends of the investigated reaches

(degraded = D; 1st restored = R1; 2nd restored = R2) of River Ruhr in summer 2014. Vertical grey bars indicate peak flow

events.




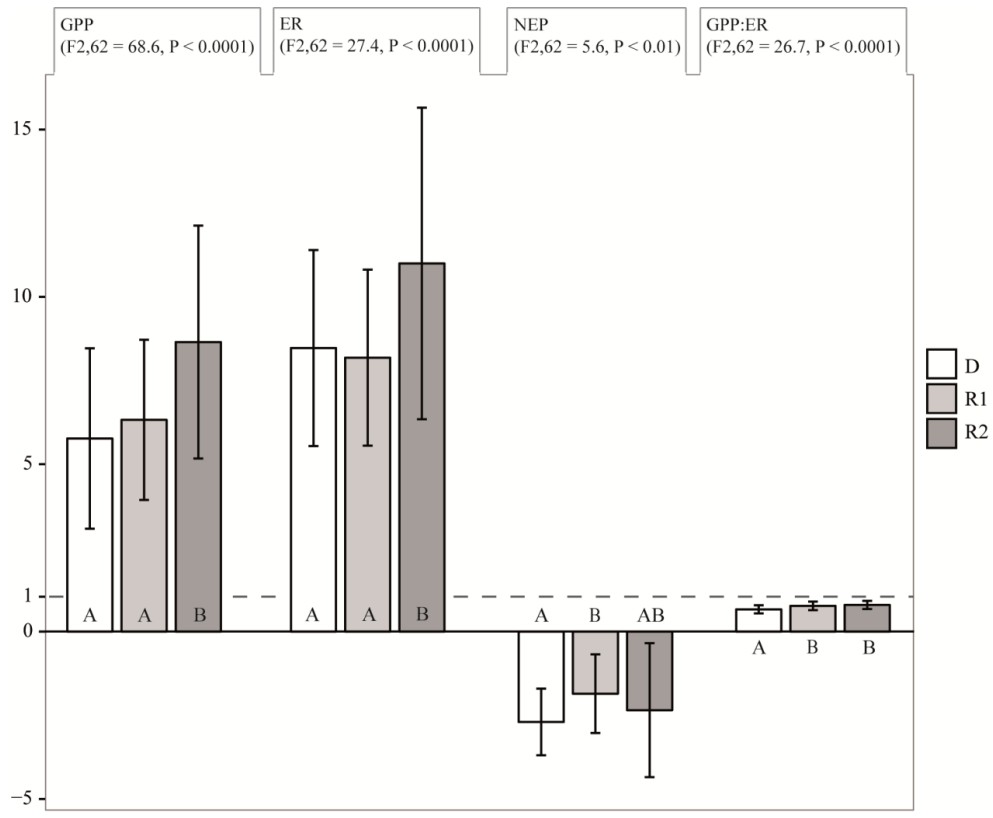

596

**Fig. 6:** Mean GPP, ER, NEP and GPP:ER ± 1SD of the sampling stations (D = station at the downstream end of the degraded reach, R1 = station at the end of 1st restored reach, R2 = station at the end of the 2nd restored reach). Results of repeated measures ANOVA in parentheses. Significant differences among stations (Tukey's HSD, $P < 0.05$) are indicated by different uppercase letters. Data of days during flow peaks were omitted from the analyses.



**Appendix S1: Information about restoration activities and restored reaches**

The restored reaches (R1 and R2) were compared to an upstream degraded "control-section". We selected the degraded reach (D) to be characteristic for the channelized state of the River Ruhr, and to reflect the conditions of the restored reaches prior to restoration (Fig. S1, S2). Accordingly, the hydromorphology of the degraded reach had been largely modified by channelization and bank fixation, resulting in lower physical stream quality (e.g. smaller wetted channel width, no islands and no accumulations of woody debris).

Restoration involved the widening of the riverbed and the reconnection of the river with its floodplain by creating a shallower river profile and by removing bank fixations. Furthermore, secondary channels and island were generated, instream structures - such as woody debris - were added and shallow habitats were created, potentially providing more space for autotrophs (Fig. S3, S4, S5, S6, S7, S8). The restored reaches differed in restoration effort (R1: moderate restoration effort and R2: high restoration effort). Briefly, R2 represented higher effort than R1 due to larger soil moving activities and higher costs for measures implemented (Table S1). Moreover, differences in restoration effort were obvious from measures implemented along the two reaches: In R1, removal of bank fixation and widening of the riverbed mainly focused on one (right) shoreline only, while the other (left) shoreline remained fixed due to railroad constrains (Fig. S7). On the contrary, R2 was substantially widened, bank fixation was removed at both shorelines and islands were created along the reach (Fig. S8). The differences between the restored reaches are further described by measurement results presented in our study (Table 2).

**Table S1:** Restoration costs and soil moving activities indicating differences in restoration effort between R1 and R2

| Reach | Costs (€) | Soil excavation (m³) | Soil shifting (m³) |
|---|---|---|---|
| **R1** | 1,400,000 | 44,000 | 15,000 |
| **R2** | 1,930,000 | 61,000 | 18,000 |





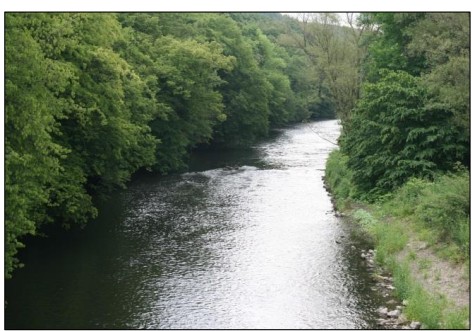

Fig. S1: Photo of the upstream degraded „control-section" (D) (photo by A. Lorenz).

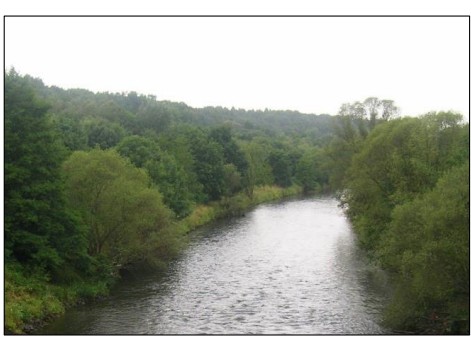

Fig. S2: Conditions of restored reaches prior to restoration (photo by A. Lorenz).

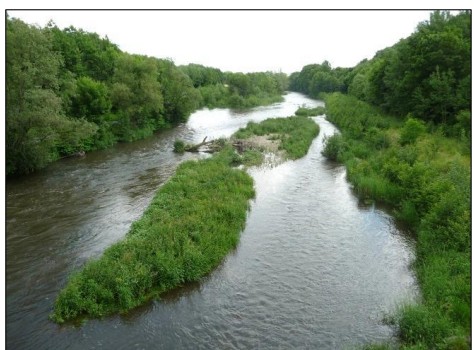

Fig. S3: Photo of the 1st restored reach (R1) (photo by B. Kupilas).

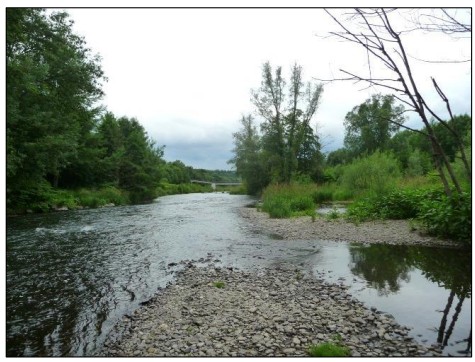

Fig. S4: Photo of the 1st restored reach (R1) (photo by B. Kupilas).

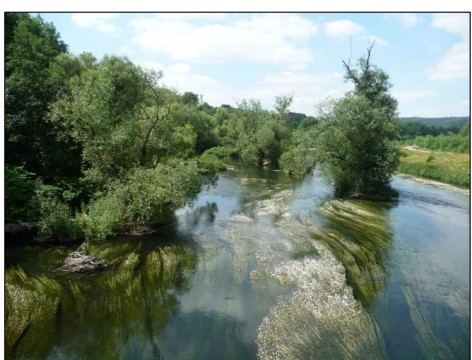

Fig. S5: Photo of the 2nd restored reach (R2) (photo by B. Kupilas).

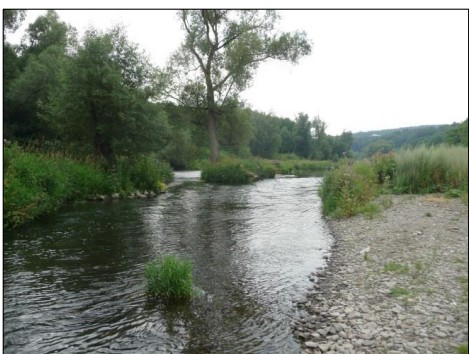

Fig. S6: Photo of the 2nd restored reach (R2) (photo by B. Kupilas).



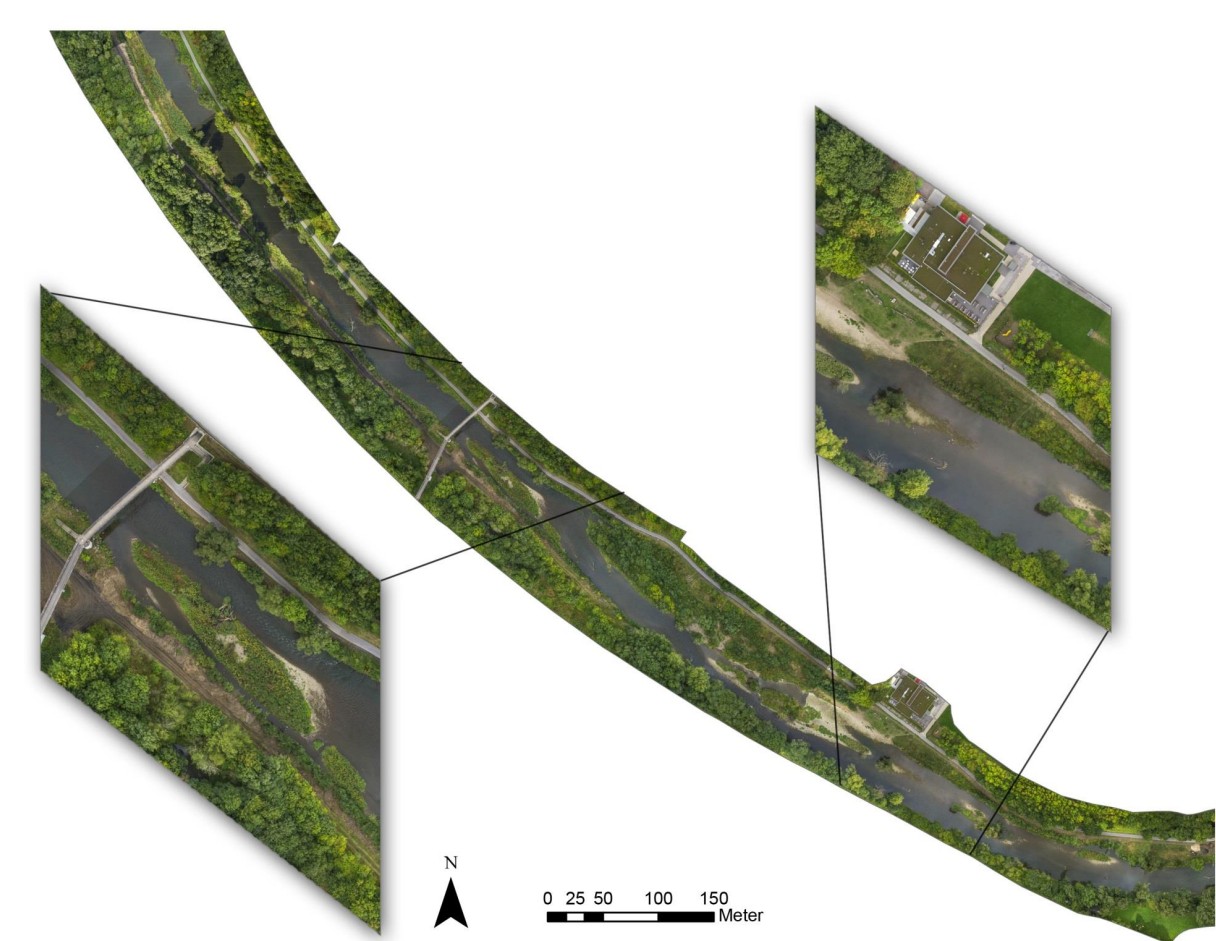

**Fig. S7:** 1st restored reach (R1) (photo by NZO GmbH, Germany).





**Fig. S8:** 2<sup>nd</sup> restored reach (R2) (photo by NZO GmbH, Germany).




**Appendix S2: $K_{oxy}^{20}$ - discharge relationships for stations in D, R1 and R2.**

**All regressions with P>0.05**

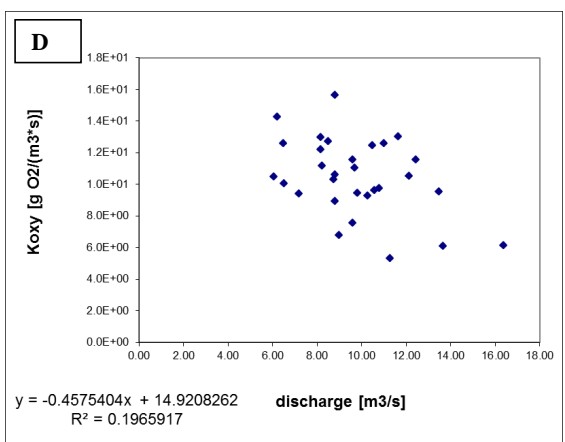

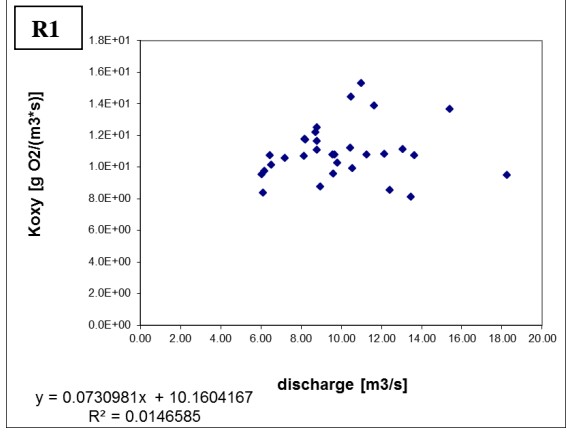

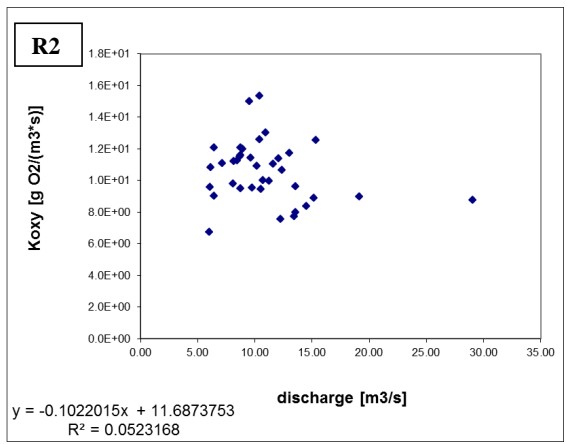





**Appendix S3: Diurnal patterns of ecosystem metabolism in the sampling stations at D, R1 and R2 for days on which GPP and ER were among the highest respectively lowest rates measured**

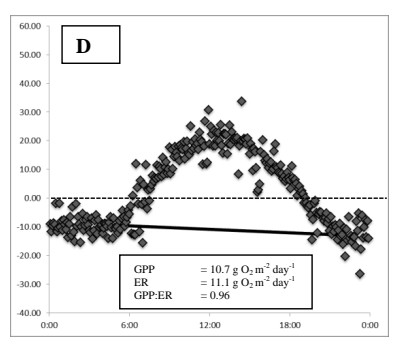
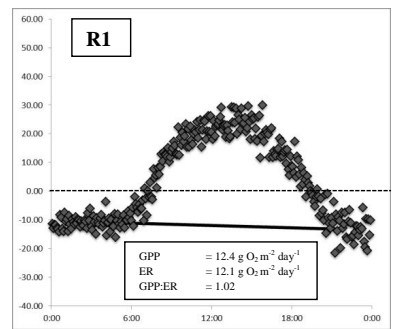
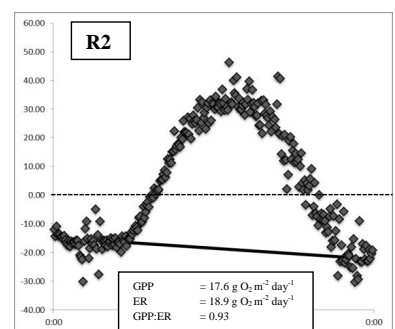

Day 17

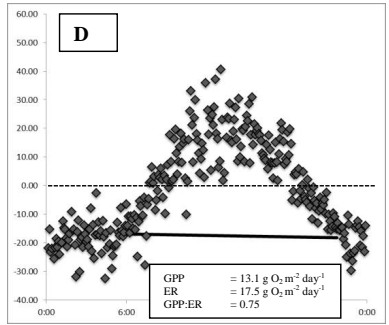
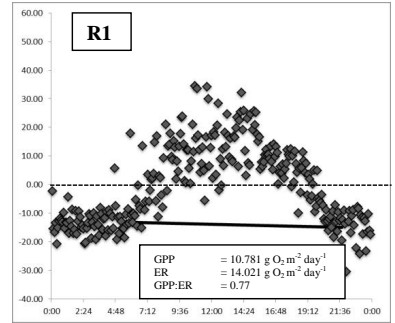
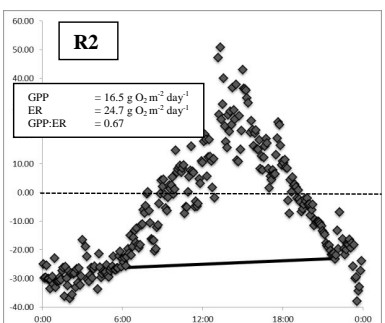





Day 1

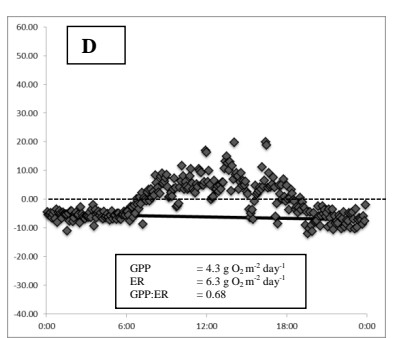

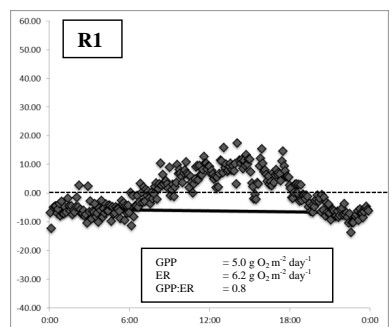

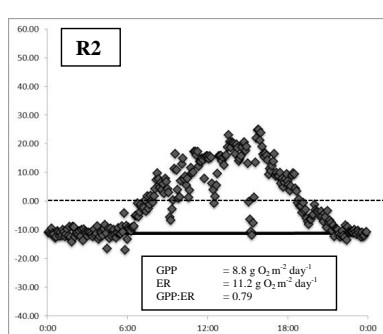

Day 40

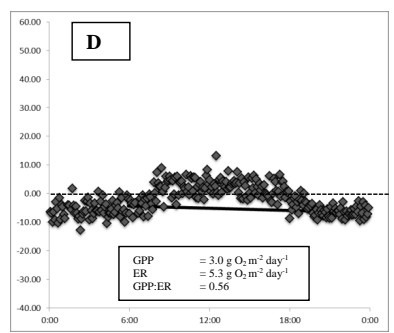

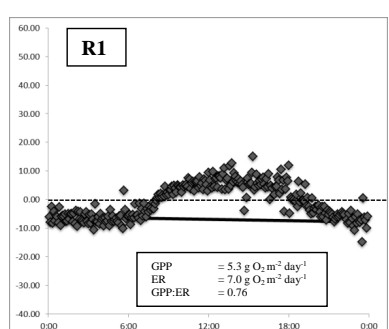

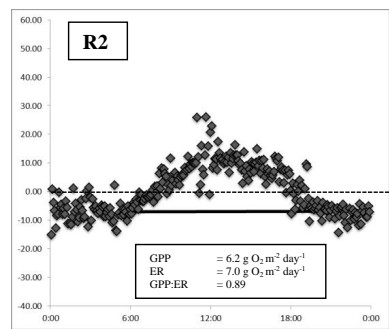





**Appendix S4: Comparison of metabolic rates estimated in our study with literature data**

GPP and ER estimated in this study were among the highest values reported for similar sized rivers (discharge between 5 - 50 m³ s⁻¹, Appendix S5); especially those of the sampling station R2. In comparison to other streams, higher GPP and ER were reported for formerly polluted streams with a channelized river course and degraded floodplain in the Basque country (Izagirre et al. 2008); accordingly, a direct comparison to the Ruhr seems inappropriate. Besides size, none of the rivers in our literature review was comparable to the Ruhr regarding the river characteristics: sediment structure, hydromorphology/river state, macrophytes, and geographic region (Appendix S5). Consequently, metabolism reference values from rivers similar to the Ruhr are not available. However, higher GPP and ER after restoration of flow patterns have been reported by Colangelo (2007), supporting our findings of higher metabolic rates following restoration. Of all the rivers for which metabolism has been reported, the channelized river Thur (Uehlinger 2006) is closest to the Ruhr regarding size, sediment, and region. Average GPP and ER reported for the Thur were similar to those of the channelized sampling station D. Thus, relatively low GPP and ER in hydromorphologically altered rivers may be common.

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





**Appendix S5: Comparison with literature data, (a) river charatersitics**

| Sampled river | River characteristics | | | | | |
|---|---|---|---|---|---|---|
| Name, geographic region | Sediment structure | Hydromorphology/river state | Macrophytes | Additional information | Width (m) | Q (m³ s⁻¹) |
| Kissimmee River, Florida, USA | Sand | Channelised, restored habitat structure in river channel with continuous flow | Reduced cover of floating and mat forming vegetation | Sub-tropical, low-gradient, blackwater | 15 – 30 | 36.60 |
| Kansas River, Kansas, USA | Sand | Slightly braided, moderatley degraded (oxbow wetlands gone, bordered by cropland, no heavy industry or large urban area, some reservoirs) | No macrophytes, diatoms main primary producers | Prairie river, shallow | 75 | 14.36 |
| Omo River, Fuji River Basin, Japan | Cobbles, boulders | Relatively good, degraded water quality due to agricultural land use | Less than 5% cover | Open-canopy lowland stream draining urban and agricultural land | N.a. | 5.12 |
| Aizarnazabal, Basque Country, Spain | Bedrock, cobble | Narrow and steep valleys with short and steep streams, biotic index: excellent | Occasionally, periphyton main primary producer | Humid-oceanic climate, formerly polluted | 22.7 | 6.27 |
| Alegia, Basque Country, Spain | Bedrock, cobble | Narrow and steep valleys with short and steep streams, biotic index: good | Occasionally, periphyton main primary producer | Humid-oceanic climate, formerly polluted | 36.2 | 6.96 |
| Altzola, Basque Country, Spain | Bedrock, cobble | Narrow and steep valleys with short and steep streams, biotic index: poor | Occasionally, periphyton main primary producer | Humid-oceanic climate, formerly polluted | 31.1 | 9.47 |
| Amorebieta, Basque Country, Spain | Bedrock, cobble | Narrow and steep valleys with short and steep streams, biotic index: very poor | Occasionally, periphyton main primary producer | Humid-oceanic climate, formerly polluted | 23.3 | 5.55 |
| Lasarte, Basque Country, Spain | Bedrock, cobble | Narrow and steep valleys with short and steep streams, biotic index: fair | Occasionally, periphyton main primary producer | Humid-oceanic climate, formerly polluted | 46.4 | 22.74 |
| Little Tennessee River, North Carolina, USA | Sand becoming a mix of bedrock, large boulders, and sand | Broad alluvial valley becoming constrained valley | N.a. | N.a. | N.a. | 12.90 |
| Thur River, Switzerland | Gravel | Channelised with stabilised banks, with reach partly being opened (i.e. removal of bank fixation) | N.a. | Alpine river | 35 | 48.70 |
| Murrumbidgee River, Darlington Point, Australia | Clay, silt with sandy bars | Degraded, but not channelized | Very little macrophytes | In an agricultural area | N.a. | 22.00 |
| Daly, Australia | Sand, gravel | Natural, about 5% of the land cleared of natural vegetation, no dams, essentially natural flow, intermittent river | Very little macrophytes | 5th - 7th order, tropical, shallow, clear water, low nutrient concentration, open canopy | N.a. | 24.00 |
| Mitchell River (MCC, upper site), Australia | Sand, bedrock | Continuous run-pool channel morphology | No macrophytes | Dry season sampled, riparian vegetation present | 32 | 27.20 |
| Buffalo Fork, Wyoming, USA | Cobble, gravel/pebble | Natural | No macrophytes | N.a. | 35.2 | 19.10 |
| Green River, Wyoming, USA | Cobble, boulder | Natural | N.a. | Below a dam | 62.5 | 25.50 |
| Salmon River, USA | Cobble, gravel | Natural | No macrophytes | N.a. | 50.5 | 25.90 |
| Tippecanoe River, Indiana, USA | Gravel, pebble with sand and fine sediment | Natural | No macrophytes | N.a. | 50.6 | 19.00 |
| Muskgeon River, Michigan, USA | Sand, silt, clay with gravel and cobbles | Natural | 9% cover | N.a. | 67 | 33.00 |
| Manistee River, Michigan, USA | Sand, silt, clay with gravel and pebble | Natural | 13% cover | N.a. | 52.5 | 36.50 |
| Bear River, Utah, USA | Sand, silt, clay | Natural morphology but hydrologically altered | No macrophytes | N.a. | 37.3 | 16.00 |



| | | | | | | |
|---|---|---|---|---|---|---|
| Green River at Ouray, Utah, USA | Sand, silt, clay | Natural | 1% cover | N.a. | 111.8 | 37.90 |
| Green River at Gray Canyon, Utah, USA | Fine sediments with gravel and cobbles | Natural | < 1% cover | N.a. | 79.1 | 41.00 |
| Chena1, Alaska, USA | N.a. | Natural flow regime, undeveloped | N.a. | Sub-arctic, clear-water river, upper catchment ~undeveloped, lower catchment with urban development | N.a. | 42.00 |
| Chena2, Alaska, USA | N.a. | Natural flow regime, undeveloped | N.a. | Sub-arctic, clear-water river, upper catchment ~undeveloped, lower catchment with urban development | N.a. | 44.50 |
| Chena3, Alaska, USA | N.a. | Natural flow regime, undeveloped | N.a. | Sub-arctic, clear-water river, upper catchment ~undeveloped, lower catchment with urban development | N.a. | 47.00 |
| Chena4, Alaska, USA | N.a. | Natural flow regime, undeveloped | N.a. | Sub-arctic, clear-water river, upper catchment ~undeveloped, lower catchment with urban development | N.a. | 47.50 |
| Ichetucknee, Florida, USA | N.a. | N.a. | N.a. | N.a. | N.a. | 8.90 |
| East Fork, Indiana, USA | N.a. | Natural | N.a. | N.a. | 47.9 | 14.00 |

N.a. = not available



**Appendix S5: comparison with literature data, (b) metabolic rates**

| Sampled river | Metabolism | | | | |
|---|---|---|---|---|---|
| Name, geographic region | GPP (g O$_2$ m$^{-2}$ d$^{-1}$) | ER (g O$_2$ m$^{-2}$ d$^{-1}$) | GPP:ER | NEP (g O$_2$ m$^{-2}$ d$^{-1}$) | Reference |
| Kissimmee River, Florida, USA | 3.95 | -9.44 | 0.42 | -5.49 | Colangelo, D.J. (2007) Response of river metabolism to restoration of flow in the Kissimmee River, Florida, U.S.A. Freshwater Biology, 52, 459–470. |
| Kansas River, Kansas, USA | 8.40 | -12.12 | 0.69 | -3.72 | Dodds, W.K., J.J. Beaulieu, J.J. Eichmiller, J.R. Fischer, N.R. Franssen, D.A. Gudder, A.S. Makinster, M.J. McCarthy, J.N. Murdock, J.M. O'Brien, J.L. Tank & R.W. Sheibley (2008) Nitrogen cycling and metabolism in the thalweg of a prairie river. Journal of Geophysical Research, 113, G04029. |
| Omo River, Fuji River Basin, Japan | 3.83 | -9.13 | 0.42 | -5.30 | Iwata, T., T. Takahashi, F. Kazama et al. (2007) Metabolic balance of streams draining urban and agricultural watersheds in central Japan. Limnology, 8, 243-250. |
| Aizarnazabal, Basque Country, Spain | 11.00 | -17.20 | 0.64 | -6.20 | Izagirre, O., U. Agirre, M. Bermejo, J. Pozo & A. Elosegi (2008) Environmental controls of whole-stream metabolism identified from continuous monitoring of Basque streams. Journal of the North American Benthological Society, 27, 252–268. |
| Alegia, Basque Country, Spain | 4.40 | -12.50 | 0.35 | -8.10 | Izagirre, O., U. Agirre, M. Bermejo, J. Pozo & A. Elosegi (2008) Environmental controls of whole-stream metabolism identified from continuous monitoring of Basque streams. Journal of the North American Benthological Society, 27, 252–268. |
| Altzola, Basque Country, Spain | 6.40 | -42.60 | 0.15 | -36.20 | Izagirre, O., U. Agirre, M. Bermejo, J. Pozo & A. Elosegi (2008) Environmental controls of whole-stream metabolism identified from continuous monitoring of Basque streams. Journal of the North American Benthological Society, 27, 252–268. |
| Amorebieta, Basque Country, Spain | 2.80 | -9.80 | 0.29 | -7.00 | Izagirre, O., U. Agirre, M. Bermejo, J. Pozo & A. Elosegi (2008) Environmental controls of whole-stream metabolism identified from continuous monitoring of Basque streams. Journal of the North American Benthological Society, 27, 252–268. |
| Lasarte, Basque Country, Spain | 6.30 | -13.50 | 0.47 | -7.20 | Izagirre, O., U. Agirre, M. Bermejo, J. Pozo & A. Elosegi (2008) Environmental controls of whole-stream metabolism identified from continuous monitoring of Basque streams. Journal of the North American Benthological Society, 27, 252–268. |
| Little Tennessee River, North Carolina, USA | 3.18 | -4.07 | 0.78 | -0.89 | McTammany, M.E., J.R. Webster, E.F. Benfield & M.A. Neatrour (2003) Longitudinal patterns of metabolism in a southern Appalachian river. Journal of the North American Benthological Society, 22, 359–370. |
| Thur River, Switzerland | 5.00 | -6.20 | 0.81 | -1.20 | Uehlinger, U. 2006. Annual cycle and inter-annual variability of gross primary production and ecosystem respiration in a floodprone river during a 15-year period. Freshwater Biology, 51, 938–950. |
| Murrumbidgee River, Darlington Point, Australia | 1.71 | -1.90 | 0.90 | -0.19 | Vink, S., M. Bormans, P.W. Ford & N.J. Grigg (2005) Quantifying ecosystem metabolism in the middle reaches of Murrumbidgee River during irrigation flow releases. Marine and Freshwater Research, 56, 227–241. |
| Daly, Australia | 2.90 | -5.34 | 0.54 | -2.44 | Townsend, S.A. & A.V. Padovan (2005) The seasonal accrual and loss of benthic algae (Spirogyra) in the Daly River, an oligotrophic river in tropical Australia. Marine and Freshwater Research, 56, 317–327. |
| Mitchell River (MCC, upper site), Australia | 2.12 | -4.47 | 0.47 | -2.35 | Hunt, R.J., T.D. Jardine, S.K. Hamilton & S.E. Bunn (2012) Temporal and spatial variation in ecosystem metabolism and food web carbon transfer in a wet-dry tropical river. Freshwater Biology, 57, 435-450. |
| Buffalo Fork, Wyoming, USA | 0.80 | -3.40 | 0.24 | -2.60 | Hall, R.O., J.L. Tank, M.A. Baker, E.J. Rosi-Marshall & E.R. Hotchkiss (2016) Metabolism, Gas Exchange, and Carbon Spiraling in Rivers. Ecosystems, 19, 73-86. |
| Green River, Wyoming, USA | 19.90 | -17.50 | 1.14 | 2.40 | Hall, R.O., J.L. Tank, M.A. Baker, E.J. Rosi-Marshall & E.R. Hotchkiss (2016) Metabolism, Gas Exchange, and Carbon Spiraling in Rivers. Ecosystems, 19, 73-86. |
| Salmon River, USA | 4.00 | -5.10 | 0.78 | -1.10 | Hall, R.O., J.L. Tank, M.A. Baker, E.J. Rosi-Marshall & E.R. Hotchkiss (2016) Metabolism, Gas Exchange, and Carbon Spiraling in Rivers. Ecosystems, 19, 73-86. |
| Tippecanoe River, Indiana, USA | 2.60 | -5.30 | 0.49 | -2.70 | Hall, R.O., J.L. Tank, M.A. Baker, E.J. Rosi-Marshall & E.R. Hotchkiss (2016) Metabolism, Gas Exchange, and Carbon Spiraling in Rivers. Ecosystems, 19, 73-86. |
| Muskgeon River, Michigan, USA | 3.00 | -4.80 | 0.63 | -1.80 | Hall, R.O., J.L. Tank, M.A. Baker, E.J. Rosi-Marshall & E.R. Hotchkiss (2016) Metabolism, Gas Exchange, and Carbon Spiraling in Rivers. Ecosystems, 19, 73-86. |
| Manistee River, Michigan, USA | 3.90 | -4.40 | 0.89 | -0.50 | Hall, R.O., J.L. Tank, M.A. Baker, E.J. Rosi-Marshall & E.R. Hotchkiss (2016) Metabolism, Gas Exchange, and Carbon Spiraling in Rivers. Ecosystems, 19, 73-86. |
| Bear River, Utah, USA | 1.10 | -1.10 | 1.00 | 0.00 | Hall, R.O., J.L. Tank, M.A. Baker, E.J. Rosi-Marshall & E.R. Hotchkiss (2016) Metabolism, Gas Exchange, and Carbon Spiraling in Rivers. Ecosystems, 19, 73-86. |



| Site | | | | | Reference |
|------|------|------|------|------|-----------|
| Green River at Ouray, Utah, USA | 1.10 | -1.20 | 0.92 | -0.10 | Hall, R.O., J.L. Tank, M.A. Baker, E.J. Rosi-Marshall & E.R. Hotchkiss (2016) Metabolism, Gas Exchange, and Carbon Spiraling in Rivers. Ecosystems, 19, 73-86. |
| Green River at Gray Canyon, Utah, USA | 0.30 | -3.00 | 0.10 | -2.70 | Hall, R.O., J.L. Tank, M.A. Baker, E.J. Rosi-Marshall & E.R. Hotchkiss (2016) Metabolism, Gas Exchange, and Carbon Spiraling in Rivers. Ecosystems, 19, 73-86. |
| Chena1, Alaska, USA | 3.25 | -8.95 | 0.36 | -5.70 | Benson, E.R., M.S. Wipfli, J.E. Clapcott & N.F. Hughes (2013) Relationships between ecosystem metabolism, benthic macroinvertebrate densities, and environmental variables in a sub-arctic Alaskan river. Hydrobiologia, 701, 189–207. |
| Chena2, Alaska, USA | 2.25 | -5.80 | 0.39 | -3.55 | Benson, E.R., M.S. Wipfli, J.E. Clapcott & N.F. Hughes (2013) Relationships between ecosystem metabolism, benthic macroinvertebrate densities, and environmental variables in a sub-arctic Alaskan river. Hydrobiologia, 701, 189–207. |
| Chena3, Alaska, USA | 1.85 | -6.10 | 0.30 | -4.25 | Benson, E.R., M.S. Wipfli, J.E. Clapcott & N.F. Hughes (2013) Relationships between ecosystem metabolism, benthic macroinvertebrate densities, and environmental variables in a sub-arctic Alaskan river. Hydrobiologia, 701, 189–207. |
| Chena4, Alaska, USA | 1.95 | -5.90 | 0.33 | -3.95 | Benson, E.R., M.S. Wipfli, J.E. Clapcott & N.F. Hughes (2013) Relationships between ecosystem metabolism, benthic macroinvertebrate densities, and environmental variables in a sub-arctic Alaskan river. Hydrobiologia, 701, 189–207. |
| Ichetucknee, Florida, USA | 10.00 | -8.50 | 1.18 | 1.50 | Heffernan, J.B. & M.J. Cohen (2010) Direct and indirect coupling of primary production and diel nitrate dynamics in a subtropical spring-fed river. Limnol. Oceanogr., 55, 677–688. |
| East Fork, Indiana, USA | 4.70 | -5.60 | 0.84 | -0.90 | Hall, R.O., J.L. Tank, M.A. Baker, E.J. Rosi-Marshall & E.R. Hotchkiss (2016) Metabolism, Gas Exchange, and Carbon Spiraling in Rivers. Ecosystems, 19, 73-86. |