# Peer review of "Restoration increases river metabolism"

_Biogeosciences, 2016_

## Short Comment (SC1) · 1 Nov 2016

Hello

Interesting and relevant study. Great to see a stream metabolism study specifically focused on restoration. I also like the discussion of the importance of macrophytes and autochthonous production to stream respiration.

One point I would like to see more complete discussion of is how the aeration rates were calculated. It would be nice to see a plot of dDO/dt = ER + K(DO_deficit) for a few nights that were considered significant. Adding confusion to the aeration rate discussion is the reported units of appendix S2 in g O2/(m3*s). Is this a typo? Why not use the same units as the text in 1/day (line 215) ?

[Figure]

Also, given the relatively low aeration rate and high productivity, why not use a parameter fitting approach to model metabolism and aeration rate? It seems a more robust approach than the night time regression method.

example: Holtgrieve, Gordon W., Daniel E. Schindler, Trevor A. Branch, and Z. Teresa A'mar. "Simultaneous Quantification of Aquatic Ecosystem Metabolism and Reaeration Using a Bayesian Statistical Model of Oxygen Dynamics." Limnology and Oceanography 55, no. 3 (May 1, 2010): 1047–63. doi:10.4319/lo.2010.55.3.1047.

Your study finds very high GPP and ER estimates compared to others. This could be a direct result of an overestimate of the aeration rate. I find that a more robust method, or convincing discussion of the aeration rate, is necessary to support these findings. If an entirely new analysis applying a parameter fitting model is perhaps infeasible, empirical values from hydrodynamics and morphology would be helpful.

Also, would it be possible to apply the two station method if you combined reaches? My rough calculations indicate a reach length of about 3000 meters would be appropriate from your hydraulics and aeration rate.

Nice work,

Robert
* * *

---

## Short Comment (SC2) · 2 Nov 2016

Dear Robert,

thank you very much for your interest in our study and your helpful comments. We agree with you that the estimation of the reaeration rate is probably the most critical step in the estimation of open-channel metabolism, and deserves further discussion. In order to present a short and concise paper, we tried not to overburden our paper with methodological details, but we are happy to discuss your suggestion (see point-by-point response in *.pdf added as supplement to this comment) and to add this information to our paper.

Please also note the supplement to this comment:

[Figure]

http://www.biogeosciences-discuss.net/bg-2016-431/bg-2016-431-SC2-supplement.pdf

[Figure]

**Supplement:**

Dear Robert,

thank you very much for your interest in our study and your helpful comments. We agree with you that the estimation of the reaeration rate is probably the most critical step in the estimation of open-channel metabolism, and deserves further discussion. In order to present a short and concise paper, we tried not to overburden our paper with methodological details, but we are happy to discuss your suggestion (see point-by-point response below) and to add this information to our paper.

COMMENT: One point I would like to see more complete discussion of is how the aeration rates were calculated. It would be nice to see a plot of $dDO/dt = ER + K(DO\_deficit)$ for a few nights that were considered significant.

RESPONSE: The nighttime regression procedure is a rather standard technique, widely used in open-channel metabolism studies (e.g., Young & Huryn, Ecol. Appl. 9: 1359-1376). This technique is particularly suitable for rivers with considerable GPP that causes considerable daytime DO increases, and stable nighttime DO plateau. It is suitable for streams with $Koxy < 0.5$ m h$^{-1}$, such as the investigated river (see Demars et al., Limnol. Oceanogr.: Methods 13, 356–374). Below we show two plots (the better fit with $p < 0.01$ and $R^2 = 0.69$ and the worse one with $p < 0.05$ and $R^2 = 0.33$) from the first two sampling weeks at station R2 to give an impression about the variability in the encountered nighttime patterns. The regression slope corresponds to the Koxy (in 1/s) in these representations. In our study, we only considered significant nighttime regressions ($p < 0.05$).

[Figure]

[Figure]

COMMENT: Adding confusion to the aeration rate discussion is the reported units of appendix S2 in g O2/(m3*s). Is this a typo? Why not use the same units as the text in 1/day (line 215)?

RESPONSE: Yes, this is a typo. The y-axis units for plots in S2 should be 1/day. Thank you!

COMMENT: Also, given the relatively low aeration rate and high productivity, why not use a parameter fitting approach to model metabolism and aeration rate? It seems a more robust approach than the nighttime regression method.

RESPONSE: Inverse modeling approaches, such as BaMM (Holtgrieve et al., Limnol. Oceanogr., 55: 1047–1063) and BASE (Grace et al., Limnol. Oceanogr.: Methods 13: 103–114) require PAR irradiance input data for the simultaneous estimation of reaeration and metabolism. Unfortunately, we do not have adequate PAR data available for the studied river. Whether inverse modeling yields more reliable reaeration estimates than the nighttime regression method (and under which circumstances?) has not been tested in the literature to our knowledge. In our opinion, the best approach would be to obtain reaeration rates from different methods (gas tracer experiments, nighttime regression, inverse modeling) in future studies. However, we believe that our reaeration estimates (ranging from about 6/d to 16/d across 3 sampling stations and 50 sampling days) are realistic estimates for the investigated river.

COMMENT: Your study finds very high GPP and ER estimates compared to others. This could be a direct result of an overestimate of the aeration rate. I find that a more robust method, or convincing discussion of the aeration rate, is necessary to support these findings. If an entirely new analysis applying a parameter fitting model is perhaps infeasible, empirical values from hydrodynamics and morphology would be helpful.

RESPONSE: To our knowledge, the nighttime regression technique has not been shown to be less robust than other methods in the literature. We see the available methods rather as complementary techniques. For example, gas tracer

approaches are considered a direct and reliable method to measure reaeration, but represent only a temporal snapshot, and can yield unreliable results if conservative tracer dilution is not adequately quantified. The nighttime regression method estimates reaeration over a longer time scale and has been found to work well in less turbulent, productive streams, but is often not very useful in turbulent, shaded headwater streams. Inverse modeling may be the most integrative approach, but measured diel DO and temperature data can sometimes be uninformative and prior information (such as field tracer studies) is needed (see Holtgrieve et al., Limnol. Oceanogr.: Methods 14: 110–113). However, if not established for a specific sampling site, empirical equations may be the most uncertain option to estimate reaeration, as results of different empirical equations vary widely (at least 40-125% error in estimates, see Demars et al., Limnol. Oceanogr.: Methods 13, 356–374). Considering two contrasting hydromorphological parameter sets from our study (20 June 2014, low discharge, 9.1 $m^3$/s; and 12 July 2014, high discharge 32.7 $m^3$/s), reaeration rates estimated by the two most commonly used empirical equations indeed varied considerably, but yielded consistently higher reaeration rates than our nighttime regressions for these days (Table 1).

**Table 1.** Comparison of reaeration coefficients estimated in this study with those estimates from empirical models for the same reaches.

| Reach | Owens 1974 (surface renewal model) $K_{oxy}^{20}$ (/d) | Tsivoglou & Neil 1976 (energy dissipation model) $K_{oxy}^{20}$ (/d) | Nightime regression $K_{oxy}^{20}$ (/d) |
|---|---|---|---|
| Low Q | | | |
| D | 29 | 42 | 11 |
| R1 | 51 | 36 | 9 |
| R2 | 17 | 19 | 7 |
| High Q | | | |
| D | 48 | 127 | 11 |
| R1 | 30 | 73 | 9 |
| R2 | 15 | 45 | 9 |

Therefore, we do not think that overestimations of reaeration rates explain the high metabolic rates in our study. As discussed in the paper, we measured river metabolism during the peak of macrophyte growth in summer. This may explain the high rates of GPP and R in our study, and consequently, lower rates may be expected in other seasons.

COMMENT: Also, would it be possible to apply the two-station method if you combined reaches? My rough calculations indicate a reach length of about 3000 meters would be appropriate from your hydraulics and aeration rate.

RESPONSE: According to Demars et al. (Limnol. Oceanogr.: Methods 13, 356–374), the 2-station method is applicable to reach lengths 0.4 v/k to 1.0 v/k. In our reach system, this range corresponds to:

| D | 3283 - 8280 m |
|---|---|
| R1 | 2765 - 6912 m |
| R2 | 1624 - 4061 m |
* * *
| R1+R2 | 2199 - 5497 m |
| all 4 reaches | 2482 - 6205 m |

Thus, we cannot evaluate separate reaches in our study, but are at the lower end of this range (slightly above 0.4 v/k) for all 4 reaches combined and for R1+R2, and can therefore possibly estimate 2-station metabolism for these combinations, at least at low flow conditions. Combining all reaches is of limited use for the aim of our paper to evaluate restoration effects, but combining restored reaches R1 and R2 may indeed provide useful information. In response to your comment, we evaluated 2-station metabolism for a few sampling days for R1+R2 and it indeed appears to work. Below please find the results of the application of the 2-station method for the first sampling day (June 20, 2014). Rates of GPP were only slightly higher for the combined reaches R1+R2 using the 2-station method than for the longer reach evaluated by the 1-station method at station R2 (that included a longer unrestored upstream section), i.e. 11.4 vs. 8.8 g DO $m^{-2}$ $d^{-1}$. Rates of R, were only slightly lower with the 2-station method, i.e. 9.6 vs. 11.2 g DO $m^{-2}$ $d^{-1}$. If consistent across our dataset, these results would further support our hypothesis of increased river metabolism due to restoration. In the revised manuscript, we will test the 2-station method systematically for the combined reaches R1+R2 for all sampling days, and report the data.

[Figure]

| Ecosystem Respiration | 9.59 | g O2/m2 d |
| Gross Primary Production | 11.36 | g O2/m2 d |
| Net Ecosystem Production | 1.76 | g O2/m2 d |
| P:R | 1.18 | |

(values on y axis in g DO m-2 d-1 /1000)

REFERENCES:

Demars et al., Limnol. Oceanogr.: Methods 13, 356–374.
Holtgrieve et al., Limnol. Oceanogr., 55: 1047–1063.

Holtgrieve et al., Limnol. Oceanogr.: Methods 14: 110–113.
Grace et al., Limnol. Oceanogr.: Methods 13: 103–114.

---

## Short Comment (SC3) · 2 Nov 2016

Thank you Benjamin for the rapid and detailed response.

You have cleared up my concerns and articulated well your reasoning for utilizing the night time regression method over other methods. It would be interesting to see what an inverse modeling technique would predict, but without adequate data, that comparison will have to wait.

Happy to hear that the two station method appears useful for the combined R1 and R2 reaches, and may add value to your study.

Cheers,

Robert

---

## Referee Comment (RC1) · Anonymous Referee #1 · 22 Nov 2016

General comments

The aim of this manuscript was to use ecosystem metabolism as a measure of ecosystem function in response to river restoration. The authors estimated ecosystem metabolism over 50 days within a mid-sized river reach that contained moderate (R1) and substantial (R2) restored reaches and a degraded (unrestored) (D) reach. The idea of using ecosystem function, especially ecosystem metabolism, as a response to river restoration is compelling, and in the direction where restoration work/research is needed and likely headed.

Using open channel diel changes in dissolved oxygen, the authors used a 1-station approach to estimate ecosystem metabolism. As both restored (R1 and R2) and unrestored reaches are in succession, the 1-station approach results in the dissolved oxygen signal integrating beyond the study reach of interest. Unfortunately, the authors cannot parse out restoration effects on ecosystem metabolism because the estimates of GPP and ER encompass all of the river reaches. If I understand correctly, the authors have the data to use a 2-station approach to estimate ecosystem metabolism. I strongly urge the authors to estimate reach specific ecosystem metabolism in order to quantitatively parse out the effects of restoration on ecosystem function.

Specific comments

L31 – Hydromorphology is introduced here and mentioned throughout, but not defined. I think it would help the readers to clearly define hydromorphology early on in the manuscript. Specifically, why or how is hydromorphology linked to ecological status (as mentioned in the next sentence).

L58 – I suggest changing 'It measures. . .' to 'Ecosystem metabolism is a measure of the production. . .'

L65 – Missing a 'have' – 'The majority of these studies 'have' focused. . .'

L66 – Dodds et al., Freshwater Science – estimated ecosystem metabolism in the Mississippi River. . . reference to this study could be included. Also – Hall et al. L&O, measure GPP in the Colorado River.

L84 – What do the authors mean by 'hydrodynamics'? Beaulieu et al. 2013 measured ecosystem metabolism in an urban stream where storm events and periods of low to now flow had a strong effect on GPP and ER.

L91 – The authors predicted that the restored reaches would result in higher biomass of primary producers. I am assuming this is due to increased light availability because of the widened channels. However, increased biomass of primary producers can also indicate eutrophication. I think within the context of this study, the authors attribute increased macrophytes, etc.. to be a positive response to restoration (increased ecosystem function). But, increased macrophytes due to eutrophication or

due to light alleviation can likely have opposite ecosystem function outcomes, I think it would be beneficial to mention or briefly discuss eutrophication versus a positive ecosystem function of increased autotrophic biomass.

L101 – km to km2

L181 – L185 – I do not follow the reasoning for not using a 2-station approach to estimate ecosystem metabolism. What was the travel time between the two stations? As mentioned within the general comments – without having a reach specific estimation of ecosystem metabolism, the authors cannot infer differences in GPP and ER to restoration efforts.

L205 – Using night-time regression is ok here. However, I do agree with one of the open comments posted for this manuscript – given the high productivity in this river, it might be possible to model GPP, ER, and k. There are several packages becoming available that may help model ecosystem metabolism from oxygen data, including a 2-station approach. One package that is currently in development is StreamMetabolizer, for instance (http://usgs-r.github.io/streamMetabolizer/) or another package from Halbedel and Buttner 2014 (Methods in Ecology and Evolution).

L225 – ANOVA with repeated measures is not appropriate for time-series data. I suggest using an analysis method that includes an auto-regressive term, perhaps generalized least squares or something similar (i.e. see Zuur, A., Ieno, E. N., Walker, N., Saveliev, A. A., & Smith, G. M. (2009). Mixed Effects Models and Extensions in Ecology with R. The Quarterly Review of Biology (Vol. 84, pp. 574–405). Springer Science & Business Media. http://doi.org/10.1086/648138)

L228 – For future analyses, the authors should justify why they would exclude days during or after storms where GPP was zero. The variation in daily rates of ecosystem metabolism within one river often exceeds or is the same of that from multiple sites (see Hall 2016, Metabolism of Streams and Rivers. In J. B. Jones & E. H. Stanley (Eds.), Stream Ecosystems in a Changing Environment (pp. 151–180). El-

sevier. http://doi.org/10.1016/B978-0-12-405890-3.00004-X). I think excluding these days, other than not being able to estimate GPP or ER (which can occur with high stream discharge events) would skew the results by artificially reducing the variance.

L266-268: Why not report ranges of NEP here? Also, I understand the utility of GPP:ER ratios – but NEP can be informative as actual flux values.

L336: Given the methodological approaches using the 1-station method of estimating metabolism along with the use of ANOVA, differences in metabolism amongst the reaches is not presently supported.

---

## Short Comment (SC4) · 25 Nov 2016

We would like to thank Anonymous Referee 1 for his helpful comments. We address these comments below in a detailed point-by-point response.

COMMENT:
The aim of this manuscript was to use ecosystem metabolism as a measure of ecosystem function in response to river restoration. The authors estimated ecosystem metabolism over 50 days within a mid-sized river reach that contained moderate (R1) and substantial (R2) restored reaches and a degraded (unrestored) (D) reach. The idea of using ecosystem function, especially ecosystem metabolism, as a response to river restoration is compelling, and in the direction where restoration work/research is needed and likely headed.
Using open channel diel changes in dissolved oxygen, the authors used a 1-station approach to estimate ecosystem metabolism. As both restored (R1 and R2) and unrestored reaches are in succession, the 1-station approach results in the dissolved oxygen signal integrating beyond the study reach of interest. Unfortunately, the authors cannot parse out restoration effects on ecosystem metabolism because the estimates of GPP and ER encompass all of the river reaches. If I understand correctly, the authors have the data to use a 2-station approach to estimate ecosystem metabolism. I strongly urge the authors to estimate reach specific ecosystem metabolism in order to quantitatively parse out the effects of restoration on ecosystem function.

RESPONSE:
We agree with the reviewer that it would be helpful to exactly quantify the metabolic rates of each of the investigated river reaches (D=degraded, R1=moderately restored, R2=intensively restored in our study). As explained in the original manuscript and in our response to Robert Pennington, these river reaches were too short to permit the application of the 2-station open-channel technique. Reach lengths chosen in our study represented typical spatial scales of river restoration practice (see lines 35/36 of the original paper). In fact, we are not aware of a single restored river section in Germany in which the 2-station open-channel method could be applied, as most sections are either too short or not homogeneous enough. We also stressed in the original manuscript that the used 1-station technique measured long upstream, degraded river reaches in addition to the river reaches of interest, and that the presented metabolic rates (measured with the 1-station method at the end of each experimental reach) and their successive increase have therefore to be considered as qualitative indicators of metabolism increase due to river restoration, rather than as the exact rates in the experimental reaches. Since the upstream river state was degraded, and calculations based on 1-station approach integrate upstream, degraded section with increasing lengths of restored sections (reaches R1 and R2) along the sampling stations, increase in metabolism can be considered as effect of restoration.

However, as the referee and Robert Pennington noted, the combined river reaches R1+R2 were long enough to permit the application of the 2-station open-channel method. Following this suggestion, we have now calculated the metabolic rates of the combined reach R1+R2 (see Fig. 1), and thereby we obtained metabolic rates that can be directly compared to metabolic rates of the upstream, degraded river (measured at station D with the 1-station open channel method). Results obtained with the 1-station and the 2-station method often agree remarkably well, and can therefore be compared (e.g., Bernot et al., 2010; Beaulieu et al., 2013).

**(a)** Restored reach (R1+R2)

[Figure]

**(b)**

Fig. 1: Daily rates of (a) gross primary production (GPP: positive values, black line) and ecosystem respiration (ER: negative values, grey lines) and (b) net ecosystem production (NEP) calculated for the total restored reach (R1+R2) of the River Ruhr in summer 2014. Vertical grey bars indicate peak flow events.

Following the referee's suggestions, we also changed our statistical approach and now use an autoregressive approach to address the autocorrelation issue of time-series data. Briefly, we use the ARIMA function in [R] to identify an ARIMA models that best represents all time series (metabolic parameters at stations D, R1, R2, and reach R1+R2), estimate average parameter predictions and 95%-confidence limits for each time series based on these models, and use F-tests to test the hypothesis of differences among time series (compare Roley et al. 2014). Using this analysis, we still find an increase in river GPP, NEP and GPP:ER along the restored river section (D to R1 and R2; estimated by the 1-station method; see Fig. 2), as previously analyzed in the original submission by repeated-measures ANOVA. However, more importantly, GPP, ER, NEP, and GPP:ER were also higher (Fig. 2) in the total restored river reach (R1+R2) than in the upstream degraded river (station D).

Thus, we could quantitatively support our previous qualitative findings by estimating metabolism with the 2-station method for the combined restored reach R1+R2 (see Fig. 1). We believe that river metabolism increases due to hydromorphological restoration are now well supported in the revised paper.

[Figure]

Fig. 2: Average predicted metabolic parameters and 95% confidence intervals of time series estimated by the 1-station open-channel-method at river stations D, R1, and R2, and by the 2-station open-channel-method for river reach R1+R2. F-tests for all variables were significant (GPP: p<0.001; ER: p<0.05; NEP: p<0.005, GPP:ER: p<0.0001). Different letters indicate differences according to Tukey's post-hoc test (p<0.05).

COMMENT:

L31 – Hydromorphology is introduced here and mentioned throughout, but not defined. I think it would help the readers to clearly define hydromorphology early on in the manuscript. Specifically, why or how is hydromorphology linked to ecological status (as mentioned in the next sentence).

RESPONSE:

We agree and will add the following definition to the revised paper: For example, the national river habitat survey in Germany, which evaluated hydromorphology using 31 parameters related to the longitudinal profile, channel development, bed structure, cross profile, bank structure and adjacent land use on the scale of 100 m long river sections, concluded that the majority of German rivers was severely degraded while achieving a 'good chemical status' (Gellert et al., 2014; UBA, 2013). As the river biota depend on certain morphological and substrate features (Beisel et al., 2000; Schröder et al., 2013) about 85% of German rivers failed to reach the 'good ecological status' demanded by the WFD (EEA, 2012).

COMMENT:
L58 – I suggest changing 'It measures …' to 'Ecosystem metabolism is a measure of the production'

RESPONSE:
We agree and will correct this in the revised paper.

COMMENT:
L65 – Missing a 'have' – 'The majority of these studies 'have' focused…'

RESPONSE:
We agree and will correct this in the revised paper.

COMMENT:
L66 – Dodds et al., Freshwater Science – estimated ecosystem metabolism in the Mississippi River reference to this study could be included. Also – Hall et al. L&O, measure GPP in the Colorado River.

RESPONSE:
We have included the mentioned studies in the revised manuscript.

COMMENT:
L84 – What do the authors mean by 'hydrodynamics'? Beaulieu et al. 2013 measured ecosystem metabolism in an urban stream where storm events and periods of low to now flow had a strong effect on GPP and ER.

RESPONSE:
We agree and will rewrite this sentence as follows: Increases in transient storage zones potentially enhance ER (Fellows et al., 2001) and nutrient processing (Valett et al., 1996; Gücker and Boëchat, 2004).

COMMENT:
L91 – The authors predicted that the restored reaches would result in higher biomass of primary producers. I am assuming this is due to increased light availability because of the widened channels. However, increased biomass of primary producers can also indicate eutrophication. I think within the context of this study, the authors attribute increased macrophytes, etc. to be a positive response to restoration (increased ecosystem function). But, increased macrophytes due to eutrophication or due to light alleviation can likely have opposite ecosystem function outcomes, I think it would be beneficial to mention or briefly discuss eutrophication versus a positive ecosystem function of increased autotrophic biomass.

RESPONSE:
We agree and will discuss this issue in more detail in the revised paper. In the present case, there are no point or diffuse sources that lead to eutrophication effects in this river section. Thus, metabolic responses should be solely due to river restoration, i.e. wider channels (-> more light availability), shallower channels (-> better habitat for macrophytes), and lower current velocities (->less hydraulic stress) as also discussed in lines 70 of the original manuscript.

COMMENT:
L101 – km to km2

RESPONSE:
In our version of the manuscript "km2" is correctly represented.

COMMENT:
L181 – L185 – I do not follow the reasoning for not using a 2-station approach to estimate ecosystem metabolism. What was the travel time between the two stations? As mentioned within the general comments – without having a reach specific estimation of ecosystem metabolism, the authors cannot infer differences in GPP and ER to restoration efforts.

RESPONSE:
We will clarify this in the revised manuscript, also giving values for travel times. In our previous response to Robert Pennington, we already presented the corresponding calculations (based on travel times and reaeration coefficients) that show that the 3 experimental reaches are to short to permit the use of the 2-station method, and we will discuss this issue in more detail in the revised manuscript. As for reach specific rates, we now present these rates for the combined restored reach R1+R2 (see previous response to the referees general comments and Figs 1 and 2). Nonetheless, we do not share the referee's view that increases in 1-station metabolism along the restored reach do not permit any conclusion about restoration effects: as mentioned in the original paper and in the response to the referee's general comments, we consider increased 1-station metabolism measured downstream of restored reaches a qualitative indicator, and the referee has not provided any argument as to why this conclusion may be flawed. As mentioned above, it is almost impossible to apply the 2-station method to restored river sections, as restored stretches (at least for larger rivers) are almost always too short or too inhomogeneous.

COMMENT:
L205 – Using night-time regression is ok here. However, I do agree with one of the open comments posted for this manuscript – given the high productivity in this river, it might be possible to model GPP, ER, and k. There are several packages becoming available that may help model ecosystem metabolism from oxygen data, including a 2-station approach. One package that is currently in development is StreamMetabolizer, for instance (http://usgs-r.github.io/streamMetabolizer/) or another package from Halbedel and Buttner 2014 (Methods in Ecology and Evolution).

RESPONSE:
We agree with the reviewer that this might be possible in the future. However, the mentioned MeCa toolbox by Halbedel and Büttner (2014) is a mere MatLab implementation of the classical method by Odum (1956) and does not allow for inverse modeling or regression approaches. We have actually used this Toolbox to model our data and results are exactly the same as those obtained by our Excel model; which should be case as we used exactly the same method and equations. Further, MeCa does not allow to model k from time series, but merely estimates k from empirical equations or a two-station propane addition experiment. As for StreamMetabolizer, we agree that this is a promising approach, but currently a tested

and stable version of this software is not available. The authors' state: "This package is in development. We are using it for our own early applications and welcome bold, flexible, resilient new users who can help us make the package better." Thus, we believe that it is too early to publish results from this software.

COMMENT:
L225 – ANOVA with repeated measures is not appropriate for time-series data. I suggest using an analysis method that includes an auto-regressive term, perhaps generalized least squares or something similar (i.e. see Zuur, A., Ieno, E. N., Walker, N., Saveliev, A. A., & Smith, G. M. (2009). Mixed Effects Models and Extensions in Ecology with R. The Quarterly Review of Biology (Vol. 84, pp. 574–405). Springer Science & Business Media. http://doi.org/10.1086/648138).

RESPONSE:
We agree and have followed the suggested paper and used auto-regressive approach (see our previous response to the reviewer's first comment, and Fig. 2).

COMMENT:
L228 – For future analyses, the authors should justify why they would exclude days during or after storms where GPP was zero. The variation in daily rates of ecosystem metabolism within one river often exceeds or is the same of that from multiple sites (see Hall 2016, Metabolism of Streams and Rivers. In J. B. Jones & E. H. Stanley (Eds.), Stream Ecosystems in a Changing Environment (pp. 151–180). Elsevier. http://doi.org/10.1016/B978-0-12-405890-3.00004-X). I think excluding these days, other than not being able to estimate GPP or ER (which can occur with high stream discharge events) would skew the results by artificially reducing the variance.

RESPONSE:
We agree that storms are an important part of environmental variability and will also mention average metabolic rates including storm measurements in the revised paper. However, GPP was not detectable during storm events, and we cannot be sure whether GPP was really zero or very low, or whether high flows may have prevented the detection of GPP. We will add this information to the revised paper.

COMMENT:
L266-268: Why not report ranges of NEP here? Also, I understand the utility of GPP:ER ratios – but NEP can be informative as actual flux values.

RESPONSE:
We agree and will present NEP here in the revised paper. Furthermore, daily NEP is presented in Fig. 5 of the original manuscript.

COMMENT:
L336: Given the methodological approaches using the 1-station method of estimating metabolism along with the use of ANOVA, differences in metabolism amongst the reaches is not presently supported.

RESPONSE:
By estimating metabolism with the 2-station method for the combined restored reach R1+R2 (see Fig. 1) and by comparing ARIMA function estimates (a) along the

restored river section and (b) between the upstream degraded river and the restored reach R1+R2 (see Fig. 2), we believe that river metabolism increases due to restoration are now well supported in the revised paper.

REFERENCES:

Beaulieu, J.J., Arango, C.P., Balz, D.A. and Shuster, W.D.: Continuous monitoring reveals multiple controls on ecosystem metabolism in a suburban stream. Freshwater Biology, 58, 918–937, 2013.

Beisel, J.-N., Usseglio-Polatera, P. and Moreteau, J.-C.: The spatial heterogeneity of a river bottom: a key factor determining macroinvertebrate communities. Hydrobiologia, 422/423, 163–171, 2000.

Bernot, M.J., Sobota, D.J., Hall, R.O., Mulholland, P.J., Dodds, W.K., Webster, J.R. et al.: Inter-regional comparison of land-use effects on stream metabolism. Freshwater Biology, 55, 1874–1890, 2010.

EEA (European Environment Agency): European Waters – Assessment of Status and Pressures. EEA Report No. 8, EEA, Copenhagen. 96 pp., 2012.

Fellows, C.S., Valett, H.M. and Dahm, C.N.: Whole-stream metabolism in two montane streams: Contribution of the hyporheic zone. Limnology and Oceanography, 46, 523–531, 2001.

Gellert, G., Pottgiesser, T. and Euler, T.: Assessment of the structural quality of streams in Germany—basic description and current status. Environmental Monitoring and Assessment, 186, 3365-3378, 2014.

Gücker, B. and Boëchat, I.G.: Stream morphology controls ammonium retention in tropical headwaters. Ecology, 85, 2818–2827, 2004.

Halbedel, S. and Büttner, O.: MeCa, a toolbox for the calculation of metabolism in heterogeneous streams. Methods in Ecology and Evolution, 5, 971–975, 2014.

Odum, H.T.: Primary production in flowing waters. Limnology and Oceanography, 2, 102–117, 1956.

Roley, S.S., Tank, J.L., Griffiths, N.A., Hall, R.O. and Davis, R.T.: The influence of floodplain restoration on whole-stream metabolism in an agricultural stream: insights from a 5-year continuous data set. Freshwater Science, 33, 1043–1059, 2014.

Schröder, M., Kiesel, J., Schattmann, A., Jähnig, S., Lorenz, A.W., Kramm, S., Keizer-Vlek, H., Rolauffs, P., Graf, W., Leitner P. and Hering, D.: Substratum associations of benthic invertebrates in lowland and mountain streams. Ecological Indicators, 30, 178-189, 2013.

UBA (Federal Environment Agency): Water Resource Management in Germany Part 1: Fundamentals. Bonn, 2013.

https://www.umweltbundesamt.de/sites/default/files/medien/378/publikationen/wawi_teil_01_englisch_barrierefrei.pdf (accessed November, 25th, 2016).

Valett, H.M., Morrice, J.A., Dahm, C.N., and Campana, M.E.: Parent lithology, surface-groundwater exchange, and nitrate retention in headwater streams. Limnology and Oceanography, 41, 333-345, 1996.

---

## Referee Comment (RC2) · Anonymous Referee #2 · 30 Nov 2016

This manuscript addresses a question of high importance – how might we use measurements of ecosystem processes to monitor the health of rivers? While there has been a push to include process measurements with other monitoring efforts (including those of restored sites), data showing how process measurements can actually be used to inform monitoring in rivers are few. Here the authors estimate ecosystem metabolism and hydromorphic characteristics of connected river reaches with (n=2, "R1" and "R2") and without (n=1, "D") a history of restoration efforts and provide evidence for higher rates of ecosystem metabolism (gross primary production and ecosystem respiration; GPP and ER) in restored versus degraded reaches. What this difference means in terms of "good" or "bad" rates remains unclear without longer-term before/after data. And while the motivation for this research is very timely and will be of broad interest, a few assumptions behind methods and presentation of results limit

the impact of the manuscript in its current form.

General comments -

1. It is not clear from the descriptions in main text and supplements why the two restored reaches were separated from one another – beyond the fact that they were perhaps separate projects? While restoration effort was indeed larger in R2 than R1, it was not drastically so, especially given the larger reach area in R2 than R1. Further, using a single-station metabolism approach to estimate GPP and ER does not honor the boundaries set by the authors in naming R1 and R2 given the overlap in O2 footprints upstream of researcher-defined reach boundaries. Based on author responses to earlier comments, it appears that combining R1 and R2 is a more suitable approach, despite losing a reach "replicate" of sorts. I hope the manuscript will be revised accordingly and the authors will confirm that combined R1-R2 rates are reasonable given what R1- and R2-only rates were.

2. Time series analyses could be a more powerful way quantify differing/strengths of controls on metabolism in R vs D reaches during the 50-day deployments. A more sophisticated analysis would enhance the contribution of the paper beyond means/ranges of GPP, ER, NEP. See Roley et al 2014 Freshwater Science for an example of this and a citation of general interest. If not used here, this is at least worth a mention for future research avenues.

3. "The importance of autochthonous production for ecosystem functioning" (19-20) is a very context-dependent statement that should be expressed with caution. What does an increase in GPP mean for ecosystem health? We see this response in the R2 reach (but D was 2nd highest in macrophytes, not R1?), and higher GPP is sometimes used as a sign of ecosystem degradation: excess primary production in response to nutrient loading. Units of N are in mg, so there may be environmental issues requiring mitigation beyond physical restoration by widening river channels. Without "pre" data for R reaches, one could argue that the restoration project provided more light + warmer

temperatures needed for enhancement of macrophytes beyond "natural" conditions.

Line-specific comments –

2 – No comma needed after "Both"

49-50 – Should include a "but see..." citation here to acknowledge that there have indeed been several previous studies even if this issue is not well-studied or settled.

53 – Delete "only"

109 – It would be useful to give time frame of/since restoration here, not only in supplement

130 – What about water chemistry? Any differences among reaches? If restoration is indeed altering nutrient retention or removal, that should be reflected in downstream concentrations.

183-4 – Vague RE: which methods as written. Why not restate and give equations for this, k, and base metabolism calculation to allow readers to better assess the methods within the manuscript itself?

225 – Good to see that the authors will update their statistical analyses in a revised manuscript.

227 – Data from flood events are one of the most exciting things we can learn about from longer time series. I urge the authors not to exclude them from their analyses.

273 – "returned"

347 – Give some numbers to support this comparison in the main text. Fine to keep the citations/table in supplement.

369 – "near-natural" is a very vague description and does not seem to be supported with data

380 – "as a functional indicator"

Table 2 – Check units. Superscripts for m didn't appear in my version of the text.

Fig 2 – Needed? There are many figures, and I didn't feel that this was needed for main text. Supplemental figures are nice.

App S5 – Possible to include Q for context? See also Genzoli & Hall 2016 FWS, Davis et al 2012 RRA, Dodds et al. 2013 FWS, Hall et al 2015 L&O.

---

## Referee Comment (RC3) · Anonymous Referee #3 · 1 Dec 2016

GENERAL COMMENTS:

This manuscript examines the response of ecosystem metabolism to river restoration by comparing ecosystem metabolism among three river reaches of a mid-size river: a degraded (unrestored) reach (D), a moderately restored (R1), and a substantially restored reach (R2). The use of ecosystem metabolism to determine the effects of river restoration is fairly novel. In that sense, the manuscript represents a relevant contribution to the challenge of incorporating measures of ecosystem functioning to river monitoring, and to river restoration in particular. The manuscript is well structured and written. In general, the results are clearly and transparently exposed, limitations indicated, and details nicely presented in the appendixes.

My main concern with this manuscript relies on the fact the sampling design of this

case study may not be the most appropriate for correctly answering the important question of whether river restoration caused a significant change in ecosystem metabolism. Ecosystem metabolism was measured in only one river, only after river restoration, and only during a certain period of the year (summer). Ideally, such a question should have been addressed by measuring in several rivers, before and after restoration (BACI design), and considering several periods of the year. None of these criteria is fulfilled, and therefore, the strength of the results and its potential extrapolation to more general responses are limited. This should be at least more explicitly acknowledged in a revised version of the manuscript.

I agree with the comments that have arisen in the open discussion regarding the use of the 1-station method. As indicated in those comments and responses, the authors should incorporate the 2-station method for the restored reaches (R1 + R2) to make their statements more robust. These limitations in the metabolism estimations together with the issue in the general sampling design (previous paragraph) and the statistically significant but relatively minor changes in metabolic fluxes in restored relative to degraded reaches, makes the conclusions of a clear effect of the restoration on ecosystem metabolism not as clear as pointed out by the authors.

I also think that the last part of the discussion could be greatly improved by making more explicit recommendations and by being more convincing about the advantages of incorporating metabolism and other functional measures to river monitoring.

SPECIFIC COMMENTS: L17-18: Unclear sentence. Rephrase. L23-24: Any hints that this is occurring at the study site? L61-64: "natural changes" and "land-use change" are confusingly used in this sentence. L79: Any reference? L89: Do you mean "contiguous" instead of "continuous"? L91: Here I miss some predictions regarding the expected differences between R1 and R2. It seems important to justify the examination of two levels of restoration. L110-115: I suggest including here information on when the restoration was done. It seems important to know how much time has passed from restoration to measurements. L138: Unclear at which flow conditions these measures

were done. L195: It seems odds that some measures were done in 2013 and other sin 2014. How may this have influenced your results?

---

## Author Comment (AC1) · 31 Jan 2017

We would like to thank the three anonymous reviewers and Robert Pennington for their helpful comments. We believe that our manuscript has been substantially improved by the suggested re-analyses and changes to the text and figures, and address these changes below in a detailed point-by-point response to the comments of reviewers 2 and 3. Our final responses to reviewer 1 and Robert Pennington have already been published in previous comments. Furthermore, we add our revised manuscript as supplement to these responses.

Detailed response to reviewer 2 COMMENT This manuscript addresses a question of high importance – how might we use measurements of ecosystem processes to monitor the health of rivers? While there has been a push to include process measurements with other monitoring efforts (including those of restored sites), data showing how process measurements can actually be used to inform monitoring in rivers are few. Here the authors estimate ecosystem metabolism and hydromorphic characteristics of connected river reaches with (n=2, "R1" and "R2") and without (n=1, "D") a history of restoration efforts and provide evidence for higher rates of ecosystem metabolism (gross primary production and ecosystem respiration; GPP and ER) in restored versus degraded reaches. What this difference means in terms of "good" or "bad" rates remains unclear without longer-term before/after data. And while the motivation for this research is very timely and will be of broad interest, a few assumptions behind methods and presentation of results limit the impact of the manuscript in its current form. RESPONSE We agree with the reviewer that the incorporation of ecosystem processes in river monitoring is of high importance and we think that our revised paper will be of broad interest. By estimating metabolism with the 2-station method for the combined restored reach R1+R2 (see Fig. 1 in response to reviewer 1) and by comparing ARIMA function estimates (a) along the restored river section and (b) between the upstream degraded river and the restored reach R1+R2 (see Fig. 2 in response to reviewer 1), we believe that river metabolism increases due to restoration are now well supported in the revised paper. As for the referee's question as to what higher rates of GPP and ER mean in terms of "good" and "bad": The restored sections (R1 and R2) have a higher structural quality according to the national river habitat survey protocol, and their better (=closer to natural conditions) morphological condition compared to the degraded "control" section has been characterized throughout the manuscript. As discussed in lines 371-374 of the original manuscript, higher rates of metabolism and the occurrence of dense macrophyte stands in restored river reaches may also increase the assimilation of dissolved nutrients (Fellows et al., 2006; Gücker et al., 2006) and the sedimentation of particulate nutrients (Schulz and Gücker, 2005), thereby positively affecting water quality. Accordingly, increased rates of GPP and ER should indicate a positive response to restoration in the case of the investigated river, as they occurred in reaches closer to local reference conditions. However, a general classification of metabolism increase as either "good" or "bad" in terms of "natural" and "unnatural" conditions is currently hardly possible as there is only limited knowledge about natural geographical gradients in river metabolism (especially for mid-sized and larger rivers) as discussed in lines 347-356 and 376-380 of the original manuscript. This highlights the importance of coupling functional measures, such as metabolism, with assessments of ecosystem structure in different biomes and ecoregions. COMMENT It is not clear from the descriptions in main text and supplements why the two restored reaches were separated from one another – beyond the fact that they were perhaps separate projects? While restoration effort was indeed larger in R2 than R1, it was not drastically so, especially given the larger reach area in R2 than R1. Further, using a single-station metabolism approach to estimate GPP and ER does not honor the boundaries set by the authors in naming R1 and R2 given the overlap in O2 footprints upstream of researcher-defined reach boundaries. Based on author responses to earlier comments, it appears that combining R1 and R2 is a more suitable approach, despite losing a reach "replicate" of sorts. I hope the manuscript will be revised accordingly and the authors will confirm that combined R1-R2 rates are reasonable given what R1- and R2-only rates were. RESPONSE The larger area of R2 compared to R1 is a consequence of restoration effort (widening of the river channel and amount of soil removed) - accordingly, it is important to separate R1 and R2 to explain the local situation with two reaches restored with different restoration effort to the reader, even if these reaches do not match the dimensions of our experiments. Restoration is most often implemented in short river stretches of approximately 1 km (as described in lines 35-36 of the original manuscript). The experimental reaches, therefore, reflected these typical spatial scales and we were able to combine the assessment of reach-scale structural characteristics with changes in metabolism. Despite the mismatch between lengths of river reaches evaluated and reaches exclusively affected by restoration, we found significant effects of reach-scale restoration on whole-river metabolism using the 1-station technique. We also stressed in the original manuscript that the used 1-station technique measured long upstream, degraded river reaches in addition to the river reaches of interest, and that the presented metabolic rates (measured with the 1-station method at the end of each experimental reach) and their successive increase have therefore to be considered as qualitative indicators of metabolism increase due to river restoration, rather than as the exact rates in the experimental reaches. In response to the reviewers' concerns, we additionally calculated the metabolic rates of the combined reach R1+R2 with the 2-station method, and obtained metabolic rates that can be directly compared to metabolic rates of the upstream, degraded river (measured at station D with the 1-station open channel method). Results obtained with the 1-station and the 2-station method often agree well (e.g., Bernot et al., 2010; Beaulieu et al., 2013). Thus, we quantitatively supported our previous qualitative findings by estimating metabolism with the 2-station method for the combined restored reach R1+R2. We would like to stress that the good agreement of results of the 1-station and the 2-station method in restored reaches, i.e. that both clearly suggested metabolism increases due to restoration, may be an important finding for agency efforts to monitor restoration outcome, because the 1-station method may be more practical for routine measurements, while the 2-station technique is often considered a research method that is too complex for such purposes. Therefore, we believe that it is important to show and compare results from both techniques in our study. COMMENT Time series analyses could be a more powerful way quantify differing/strengths of controls on metabolism in R vs D reaches during the 50-day deployments. A more sophisticated analysis would enhance the contribution of the paper beyond means/ranges of GPP, ER, NEP. See Roley et al 2014 Freshwater Science for an example of this and a citation of general interest. If not used here, this is at least worth a mention for future research avenues. RESPONSE We changed our statistical approach and now use an autoregressive approach to address the autocorrelation issue of time-series data (see response to reviewer 1 for a more detailed description). However, this new approach only resulted in minor changes in results. COMMENT The importance of autochthonous production for ecosystem functioning" (19-20) is a very context-dependent statement that should be expressed with caution. What does an increase in GPP mean for ecosystem health? We see this response in the R2 reach (but D was 2nd highest in macrophytes, not R1?), and higher GPP is sometimes used as a sign of ecosystem degradation: excess primary production in response to nutrient loading. Units of N are in mg, so there may be environmental issues requiring mitigation beyond physical restoration by widening river channels. Without "pre" data for R reaches, one could argue that the restoration project provided more light + warmer temperatures needed for enhancement of macrophytes beyond "natural" conditions. RESPONSE We express increased abundance of macrophytes as positive effect of restoration (see Lorenz et al. 2012 J. Appl. Ecol.) rather than using it as indicator for high nutrient load and eutrophication. Restoration aimed to establish near-natural hydromorphology and biota. Thus, increased occurrence of macrophytes is positive effect in terms of near-natural state of the investigated river. See also response to comment of reviewer 1: We agree and discussed this issue in more detail in the revised paper. In the present case, there are no point or diffuse sources that lead to eutrophication effects in this river section. Thus, metabolic responses should be solely due to river restoration, i.e. wider channels (-> more light availability), shallower channels (-> better habitat for macrophytes), and lower current velocities (->less hydraulic stress) as also discussed in lines 70 of the original manuscript. In R1 other autotrophs (e.g., periphyton) which have not been subject to mapping may have stimulated metabolism. COMMENT L2 - No comma needed after "Both" RESPONSE We agree and corrected this in the revised paper. COMMENT L - 49-50 – Should include a "but see: : :" citation here to acknowledge that there have indeed been several previous studies even if this issue is not well-studied or settled. RESPONSE We have added this to lines 53ff of the original manuscript. COMMENT L - 53 – Delete "only" RESPONSE We corrected this in the revised paper. COMMENT L - 109 – It would be useful to give time frame of/since restoration here, not only in supplement. RESPONSE We agree and added this information in the next paragraph of the revised paper. Please see also response to reviewer 3. COMMENT L - 130 – What about water chemistry? Any differences among reaches? If restoration is indeed altering nutrient retention or removal, that should be reflected in downstream concen-

trations. RESPONSE It may well be possible that changes nutrient concentrations are measurable along the >2 km of restored river. However, we did not measure nutrient concentrations along the studied stream section, and therefore cannot comment on that. COMMENT L - 183-4 – Vague RE: which methods as written. Why not restate and give equations for this, k, and base metabolism calculation to allow readers to better assess the methods within the manuscript itself? RESPONSE We agree with the reviewer that it is interesting to have the mentioned equations and added them to the manuscript. COMMENT L - 225 – Good to see that the authors will update their statistical analyses in a revised manuscript. RESPONSE The detailed results from these analyses are in our response to reviewer 1. COMMENT L - 227 – Data from flood events are one of the most exciting things we can learn about from longer time series. I urge the authors not to exclude them from their analyses. RESPONSE We agree that floods and storms are an important part of environmental variability. However, GPP was not detectable during storm events, and we cannot be sure whether GPP was really zero or very low, or whether high flows may have prevented the detection of GPP. As reviewer 1 also stated, this often occurs with high stream discharge events. Thus, we show the data and discuss them in terms of environmental variability, but believe that it is more adequate not to analyze flood data statistically in terms of restoration effect. COMMENT L - 273 – "returned" RESPONSE We agree and corrected this in the revised paper. COMMENT L - 347 – Give some numbers to support this comparison in the main text. Fine to keep the citations/table in supplement. RESPONSE We agree and added more detailed information in the discussion. COMMENT L - 369 – "near-natural" is a very vague description and does not seem to be supported with data. RESPONSE We removed the term "near-natural" conditions ("Thus, the restoration of short river reaches may have positive effects..."). COMMENT L - 380 – "as a functional indicator". RESPONSE We agree and corrected this in the revised paper. COMMENT Table 2 – Check units. Superscripts for m didn't appear in my version of the text. RESPONSE In our version of the manuscript superscripts are correctly represented. We confirmed this in the final version. COMMENT Fig 2 – Needed? There are many figures, and I didn't feel that this was needed for main text. Supplemental figures are nice. RESPONSE We believe that these figures are important to understand our rationale. Moreover, as Biogeosciences is an online journal without figure limits, we thought that this would not be a problem. COMMENT App S5 – Possible to include Q for context? See also Genzoli & Hall 2016 FWS, Davis et al 2012 RRA, Dodds et al. 2013 FWS, Hall et al 2015 L&O. RESPONSE Yes, we included Q in spreadsheet (a) "river characteristics" of Appendix 5.

Detailed response to reviewer 3 COMMENT This manuscript examines the response of ecosystem metabolism to river restoration by comparing ecosystem metabolism among three river reaches of a mid-size river: a degraded (unrestored) reach (D), a moderately restored (R1), and a substantially restored reach (R2). The use of ecosystem metabolism to determine the effects of river restoration is fairly novel. In that sense, the manuscript represents a relevant contribution to the challenge of incorporating measures of ecosystem functioning to river monitoring, and to river restoration in particular. The manuscript is well structured and written. In general, the results are clearly and transparently exposed, limitations indicated, and details nicely presented in the appendixes. My main concern with this manuscript relies on the fact the sampling design of this case study may not be the most appropriate for correctly answering the important question of whether river restoration caused a significant change in ecosystem metabolism. Ecosystem metabolism was measured in only one river, only after river restoration, and only during a certain period of the year (summer). Ideally, such a question should have been addressed by measuring in several rivers, before and after restoration (BACI design), and considering several periods of the year. None of these criteria is fulfilled, and therefore, the strength of the results and its potential extrapolation to more general responses are limited. This should be at least more explicitly acknowledged in a revised version of the manuscript. RESPONSE The comparison of restored with upstream degraded "control" sections (space-for-time substitution) is commonly used to quantify restoration effects as data on pre-restoration are rare (e.g., Hering et al. 2015 J. Appl. Ecol., Jähnig et al. 2011 J. Appl. Ecol.). Time of measure-

ments is in accordance with the WFD compliant sampling period for structural measures such as macroinvertebrates and macrophytes. Consequently, our results can be linked to these measures (see lines 315-318). The measurements are extensive and time consuming and it was not the aim of our paper to replicate measurements in several rivers – this would have required a much simpler method, thus gaining limited insights into the underlying mechanisms. We would like to stress that our manuscript is the first measuring metabolic changes of medium-sized rivers following restoration. The lack of available data - especially for mid-sized and larger rivers - is discussed in lines 351-356 of the original manuscript.

COMMENT I agree with the comments that have arisen in the open discussion regarding the use of the 1-station method. As indicated in those comments and responses, the authors should incorporate the 2-station method for the restored reaches (R1 + R2) to make their statements more robust. These limitations in the metabolism estimations together with the issue in the general sampling design (previous paragraph) and the statistically significant but relatively minor changes in metabolic fluxes in restored relative to degraded reaches, makes the conclusions of a clear effect of the restoration on ecosystem metabolism not as clear as pointed out by the authors. RESPONSE Please see our response to reviewer 1. We do not consider significant GPP increases from ≈5 to ≈7 g DO m-2 d-1 and significant NEP increases from ≈-2 to ≈0 g DO m-2 d-1 as minor changes from an ecosystem perspective.

COMMENT I also think that the last part of the discussion could be greatly improved by making more explicit recommendations and by being more convincing about the advantages of incorporating metabolism and other functional measures to river monitoring.

RESPONSE We agree with the reviewer that advantages of functional indicators for river monitoring are manifold and should be mentioned in studies, which are using functional measures in applied river research. However, we feel that this topic - and ecosystem metabolism in particular - has been widely discussed within the literature (see also references presented in line 382 of the original manuscript) and that a summary of the benefits of metabolism as a functional indicator is more appropriate as presented in lines 380 - 387 of the original manuscript. We think that a more extensive description would make the manuscript lengthier without giving new information.

COMMENT L - 17-18: Unclear sentence. Rephrase.

RESPONSE We changed the sentence as follows: "Restoration increased autotrophic processes, as indicated by higher GPP:ER rates measured at restored reaches".

COMMENT L - 23-24: Any hints that this is occurring at the study site?

RESPONSE "High rates of metabolism and the occurrence of dense macrophyte stands may increase the assimilation of dissolved nutrients and the sedimentation of particulate nutrients, thereby positively affecting water quality." – This is our interpretation of possible consequences of the observed high metabolism rates, although we did not measure nutrients permanently. To highlight the speculative nature of this sentence we use the word "may".

COMMENT L - 61-64: "natural changes" and "land-use change" are confusingly used in this sentence.

RESPONSE We think the difference is clearly described by giving examples in the original manuscript (see lines 62 and 64). Natural changes refer to floods and droughts (e.g., Uehlinger, 2000) and land-use change refers to differences between pristine and agricultural streams (e.g., Gücker et al., 2009; Silva-Junior et al., 2014).

COMMENT L - 79: Any reference?

RESPONSE "Moreover, the reconnection of rivers with their floodplains by creating shallower river profiles and removing bank fixations may enhance inundation frequency, and hence resource transfers from land to water" - This is our interpretation of possible consequences of the hydromorphological restoration (reconnection of river and floodplain by creating shallower river profiles and removing bank fixations) al-

though we are not aware of a reference. To highlight the speculative nature of this sentence we use the word "may".

COMMENT L - 89: Do you mean "contiguous" instead of "continuous"?

RESPONSE We agree and corrected this in the revised paper. COMMENT L - 91: Here I miss some predictions regarding the expected differences between R1 and R2. It seems important to justify the examination of two levels of restoration.

RESPONSE We expected (i) hydromorphological river characteristics, i.e. habitat composition and hydrodynamics, to change following restoration, with the magnitude of change depending on restoration effort (e.g. width and diversity of the river channel, and abundance of primary producers, as well as sizes and locations of transient storage zones in the two restored river reaches compared to the degraded reach).

COMMENT L - 110-115: I suggest including here information on when the restoration was done. It seems important to know how much time has passed from restoration to measurements.

RESPONSE We agree and added this information in the revised paper. See also response to reviewer 2. COMMENT L - 138: Unclear at which flow conditions these measures were done. RESPONSE We agree and added this information in the revised paper.

COMMENT L - 195: It seems odds that some measures were done in 2013 and others in 2014. How may this have influenced your results?

RESPONSE The purpose of our study was to quantify structural changes following restoration at the reach scale and examine the related effects on metabolism. Considering that there were no major meteorological differences between these years, that structural differences between D, R1 and R2 (effort) were very similar between years, and that measurements were performed in the same season at similar discharge conditions, we believe that interannual variability is a minor confounding factor in our study.

Please also note the supplement to this comment:
http://www.biogeosciences-discuss.net/bg-2016-431/bg-2016-431-AC1-supplement.pdf

―――――――――――――――――――――

[Figure]

**Supplement:**

[revised manuscript text omitted]

---

## Author Response (AR1)

**Response to the reviewers' comments**

**Summary**

We would like to thank the three anonymous reviewers and Robert Pennington for their helpful comments during the interactive discussion stage. Following the reviewers' suggestions carefully, we have performed major re-analyses of our data and changes to the text and figures. We believe that our manuscript has been substantially improved by this major revision, and address these changes below in a detailed point-by-point response to the comments of the reviewers and Robert Pennington. Most relevant changes involved the additional calculation of metabolic rates for the combined restored reach (R1+R2) using the two-station method. Further, we changed our statistical approach and now use an autoregressive approach to address the autocorrelation issue of time-series data (using the ARIMA function in [R]). While the results of this new, statistically sophisticated approach closely resemble those of previously used simple rmANOVA, the use of the 2-station approach quantitatively supported our previous qualitative findings obtained from the one-station method. We believe that the use of the 2-station method was the most substantial criticism and that the revised manuscript addresses this issue well by exactly providing the data the reviewers asked for.

We would also like to emphasize the good agreement of results of the one-station and the two-station method in our restored reaches, i.e. that both clearly suggested metabolism increases due to restoration. This is an important finding from a practical point of view, e.g. for agency efforts to monitor restoration outcome, because the 1-station method is often more practical for routine measurements, while the 2-station technique is often considered a research method that is too complex for such purposes. Therefore, we believe that it is important to show and compare results from both techniques side by side in our study. At the end of this document, we also added our revised manuscript with changes marked as supplement to these responses.

**Detailed response to reviewer 1**

COMMENT:
The aim of this manuscript was to use ecosystem metabolism as a measure of ecosystem function in response to river restoration. The authors estimated ecosystem metabolism over 50 days within a mid-sized river reach that contained moderate (R1) and substantial (R2) restored reaches and a degraded (unrestored) (D) reach. The idea of using ecosystem function, especially ecosystem metabolism, as a response to river restoration is compelling, and in the direction where restoration work/research is needed and likely headed.

Using open channel diel changes in dissolved oxygen, the authors used a 1-station approach to estimate ecosystem metabolism. As both restored (R1 and R2) and unrestored reaches are in succession, the 1-station approach results in the dissolved oxygen signal integrating beyond the study reach of interest. Unfortunately, the authors cannot parse out restoration effects on ecosystem metabolism because the estimates of GPP and ER encompass all of the river reaches. If I understand correctly, the authors have the data to use a 2-station approach to estimate ecosystem metabolism. I strongly urge the authors to estimate reach specific ecosystem metabolism in order to quantitatively parse out the effects of restoration on ecosystem function.

RESPONSE:

We agree with the reviewer that it would be helpful to exactly quantify the metabolic rates of each of the investigated river reaches (D=degraded, R1=moderately restored, R2=intensively restored in our study). As explained in the original manuscript and in our response to Robert Pennington, these river reaches were too short to permit the application of the 2-station open-channel technique. Reach lengths chosen in our study represented typical spatial scales of river restoration practice (see lines 35/36 of the original paper). In fact, we are not aware of a single restored river section in Germany in which the 2-station open-channel method could be applied, as most sections are either too short or not homogeneous enough. We also stressed in the original manuscript that the used 1-station technique measured long upstream, degraded river reaches in addition to the river reaches of interest, and that the presented metabolic rates (measured with the 1-station method at the end of each experimental reach) and their successive increase have therefore to be considered as qualitative indicators of metabolism increase due to river restoration, rather than as the exact rates in the experimental reaches. Since the upstream river state was degraded, and calculations based on 1-station approach integrate upstream, degraded section with increasing lengths of restored sections (reaches R1 and R2) along the sampling stations, increase in metabolism can be considered as effect of restoration.

However, as the referee and Robert Pennington noted, the combined river reaches R1+R2 were long enough to permit the application of the 2-station open-channel method. Following this suggestion, we have now calculated the metabolic rates of the combined reach R1+R2 (see Fig. 1), and thereby we obtained metabolic rates that can be directly compared to metabolic rates of the upstream, degraded river (measured at station D with the 1-station open channel method). Results obtained with the 1-station and the 2-station method often agree remarkably well, and can therefore be compared (e.g., Bernot et al., 2010; Beaulieu et al., 2013).

[Figure]

**(a)** Restored reach (R1+R2)

**(b)**

Fig. 1: Daily rates of (a) gross primary production (GPP: positive values, black line) and ecosystem respiration (ER: negative values, grey lines) and (b) net ecosystem production (NEP) calculated for the total restored reach (R1+R2) of the River Ruhr in summer 2014. Vertical grey bars indicate peak flow events.

Following the referee's suggestions, we also changed our statistical approach and now use an autoregressive approach to address the autocorrelation issue of time-series data. Briefly, we use the ARIMA function in [R] to identify an ARIMA model that best represents all time series (metabolic parameters at stations D, R1, R2, and reach R1+R2), estimate average parameter predictions and 95%-confidence limits for each time series based on these models, and use F-tests to test the hypothesis of differences among time series (compare Roley et al. 2014). Using this analysis, we still find an increase in river GPP, NEP and GPP:ER along the restored river section (D to R1 and R2; estimated by the 1-station method; see Fig. 2), as previously analyzed in the original submission by repeated-measures ANOVA. However, more importantly, GPP, ER, NEP, and GPP:ER were also higher (Fig. 2) in the total restored river reach (R1+R2) than in the upstream degraded river (station D).

Thus, we could quantitatively support our previous qualitative findings by estimating metabolism with the 2-station method for the combined restored reach R1+R2 (see Fig. 1). We believe that river metabolism increases due to hydromorphological restoration are now well supported in the revised paper.

[Figure]

Fig. 2: Average predicted metabolic parameters and 95% confidence intervals of time series estimated by the 1-station open-channel-method at river stations D, R1, and R2, and by the 2-station open-channel-method for river reach R1+R2. F-tests for all variables were significant (GPP: $p<0.001$; ER: $p<0.05$; NEP: $p<0.005$, GPP:ER: $p<0.0001$). Different letters indicate differences according to Tukey's post-hoc test ($p<0.05$).

COMMENT:
L31 – Hydromorphology is introduced here and mentioned throughout, but not defined. I think it would help the readers to clearly define hydromorphology early on in the manuscript. Specifically, why or how is hydromorphology linked to ecological status (as mentioned in the next sentence).

RESPONSE:
We agree and added the following definition to the revised paper: For example, the German national river habitat survey, which evaluates 31 hydromorphological parameters for 100 m river sections, concluded that the majority of German rivers is severely degraded (Gellert et al., 2014; UBA, 2013). As the river biota depend on suited habitats (Beisel et al., 2000; Schröder et al., 2013), about 85% of German rivers failed to reach the 'good ecological status' demanded by the WFD (EEA, 2012).

COMMENT:
L58 – I suggest changing 'It measures …' to 'Ecosystem metabolism is a measure of the production'

RESPONSE:
We agree and corrected this in the revised paper.

COMMENT:
L65 – Missing a 'have' – 'The majority of these studies 'have' focused…'

RESPONSE:
We agree and corrected this in the revised paper.

COMMENT:
L66 – Dodds et al., Freshwater Science – estimated ecosystem metabolism in the Mississippi River reference to this study could be included. Also – Hall et al. L&O, measure GPP in the Colorado River.

RESPONSE:
We have included the mentioned studies in the revised manuscript.

COMMENT:
L84 – What do the authors mean by 'hydrodynamics'? Beaulieu et al. 2013 measured ecosystem metabolism in an urban stream where storm events and periods of low to now flow had a strong effect on GPP and ER.

RESPONSE:
We agree and rewrote this sentence as follows: Increases in transient storage zones potentially enhance ER (Fellows et al., 2001) and nutrient processing (Valett et al., 1996; Gücker and Boëchat, 2004).

COMMENT:
L91 – The authors predicted that the restored reaches would result in higher biomass of primary producers. I am assuming this is due to increased light availability because of the widened channels. However, increased biomass of primary producers can also indicate eutrophication. I think within the context of this study, the authors attribute increased macrophytes, etc. to be a positive response to restoration (increased ecosystem function). But, increased macrophytes due to eutrophication or due to light alleviation can likely have opposite ecosystem function outcomes, I think it would be beneficial to mention or briefly discuss eutrophication versus a positive ecosystem function of increased autotrophic biomass.

RESPONSE:
We agree and discussed this issue in more detail in the revised paper. In the present case, there are no point or diffuse sources that lead to eutrophication effects in this river section. Thus, metabolic responses should be solely due to river restoration, i.e. wider channels (-> more light availability), shallower channels (-> better habitat for macrophytes), and lower current velocities (->less hydraulic stress) as also discussed in lines 70 of the original manuscript.

COMMENT:
L101 – km to km2

RESPONSE:
In our version of the manuscript "km2" is correctly represented.

COMMENT:
L181 – L185 – I do not follow the reasoning for not using a 2-station approach to estimate ecosystem metabolism. What was the travel time between the two stations? As mentioned within the general comments – without having a reach specific estimation of ecosystem metabolism, the authors cannot infer differences in GPP and ER to restoration efforts.

RESPONSE:
We clarified this in the revised manuscript, also giving values for travel times. In our response to Robert Pennington, we already presented the corresponding calculations (based on travel times and reaeration coefficients) that show that the 3 experimental reaches are too short to permit the use of the 2-station method, and we discussed this issue in more detail in the revised manuscript. As for reach specific rates, we now present these rates for the combined restored reach R1+R2 (see previous response to the referees general comments and Figs 1 and 2). Nonetheless, we do not share the referee's view that increases in 1-station metabolism along the restored reach do not permit any conclusion about restoration effects: as mentioned in the original paper and in the response to the referee's general comments, we consider increased 1-station metabolism measured downstream of restored reaches a qualitative indicator, and the referee has not provided any argument as to why this conclusion may be flawed. As mentioned above, it is almost impossible to apply the 2-station method to restored river sections, as restored stretches (at least for larger rivers) are almost always too short or too inhomogeneous.

COMMENT:
L205 – Using night-time regression is ok here. However, I do agree with one of the open comments posted for this manuscript – given the high productivity in this river, it might be possible to model GPP, ER, and k. There are several packages becoming available that may help model ecosystem metabolism from oxygen data, including a 2-station approach. One package that is currently in development is StreamMetabolizer, for instance (http://usgs-r.github.io/streamMetabolizer/) or another package from Halbedel and Buttner 2014 (Methods in Ecology and Evolution).

RESPONSE:
We agree with the reviewer that this might be possible in the future. However, the mentioned MeCa toolbox by Halbedel and Büttner (2014) is a mere MatLab implementation of the classical method by Odum (1956) and does not allow for inverse modeling or regression approaches. We have actually used this Toolbox to model our data and results are exactly the same as those obtained by our Excel model; which should be case as we used exactly the same method and equations. Further, MeCa does not allow to model k from time series, but merely estimates k from empirical equations or a two-station propane addition experiment. As for StreamMetabolizer, we agree that this is a promising approach, but currently a tested and stable version of this software is not available. The authors' state: "This package is in development. We are using it for our own early applications and welcome bold, flexible, resilient new users who can help us make the package better." Thus, we believe that it is too early to publish results from this software.

COMMENT:
L225 – ANOVA with repeated measures is not appropriate for time-series data. I suggest using an analysis method that includes an auto-regressive term, perhaps generalized least squares or something similar (i.e. see Zuur, A., Ieno, E. N., Walker, N., Saveliev, A. A., & Smith, G. M. (2009). Mixed Effects Models and Extensions in Ecology with R. The Quarterly Review of Biology (Vol. 84, pp. 574–405). Springer Science & Business Media. http://doi.org/10.1086/648138).

RESPONSE:
We agree and have followed the suggested paper and used auto-regressive approach (see our previous response to the reviewer's first comment, and Fig. 2).

COMMENT:
L228 – For future analyses, the authors should justify why they would exclude days during or after storms where GPP was zero. The variation in daily rates of ecosystem metabolism within one river often exceeds or is the same of that from multiple sites (see Hall 2016, Metabolism of Streams and Rivers. In J. B. Jones & E. H. Stanley (Eds.), Stream Ecosystems in a Changing Environment (pp. 151–180). Elsevier. http://doi.org/10.1016/B978-0-12-405890-3.00004-X). I think excluding these days, other than not being able to estimate GPP or ER (which can occur with high stream discharge events) would skew the results by artificially reducing the variance.

RESPONSE:
We agree that storms are an important part of environmental variability and mentioned average metabolic rates including storm measurements in the revised paper. However, GPP was not detectable during storm events, and we cannot be sure whether GPP was really zero or very low, or whether high flows may have prevented the detection of GPP. We will add this information to the revised paper.

COMMENT:
L266-268: Why not report ranges of NEP here? Also, I understand the utility of GPP:ER ratios – but NEP can be informative as actual flux values.

RESPONSE:
We agree and presented NEP here in the revised paper. Furthermore, daily NEP is presented in Fig. 6 of the revised manuscript.

COMMENT:
L336: Given the methodological approaches using the 1-station method of estimating metabolism along with the use of ANOVA, differences in metabolism amongst the reaches is not presently supported.

RESPONSE:
By estimating metabolism with the 2-station method for the combined restored reach R1+R2 (see Fig. 1) and by comparing ARIMA function estimates (a) along the restored river section and (b) between the upstream degraded river and the restored reach R1+R2 (see Fig. 2), we believe that river metabolism increases due to restoration are now well supported in the revised paper.

COMMENT

It is not clear from the descriptions in main text and supplements why the two restored reaches were separated from one another – beyond the fact that they were perhaps separate projects? While restoration effort was indeed larger in R2 than R1, it was not drastically so, especially given the larger reach area in R2 than R1. Further, using a single-station metabolism approach to estimate GPP and ER does not honor the boundaries set by the authors in naming R1 and R2 given the overlap in O2 footprints upstream of researcher-defined reach boundaries. Based on author responses to earlier comments, it appears that combining R1 and R2 is a more suitable approach, despite losing a reach "replicate" of sorts. I hope the manuscript will be revised accordingly and the authors will confirm that combined R1-R2 rates are reasonable given what R1- and R2-only rates were.

RESPONSE

The larger area of R2 compared to R1 is a consequence of restoration effort (widening of the river channel and amount of soil removed) - accordingly, it is important to separate R1 and R2 to explain the local situation with two reaches restored with different restoration effort to the reader, even if these reaches do not match the dimensions of our experiments.

Restoration is most often implemented in short river stretches of approximately 1 km (as described in lines 35-36 of the original manuscript). The experimental reaches, therefore, reflected these typical spatial scales and we were able to combine the assessment of reach-scale structural characteristics with changes in metabolism. Despite the mismatch between lengths of river reaches evaluated and reaches exclusively affected by restoration, we found significant effects of reach-scale restoration on whole-river metabolism using the 1-station technique. We also stressed in the original manuscript that the used 1-station technique measured long upstream, degraded river reaches in addition to the river reaches of interest, and that the presented metabolic rates (measured with the 1-station method at the end of each experimental reach) and their successive increase have therefore to be considered as qualitative indicators of metabolism increase due to river restoration, rather than as the exact rates in the experimental reaches.

In response to the reviewers' concerns, we additionally calculated the metabolic rates of the combined reach R1+R2 with the 2-station method, and obtained metabolic rates that can be directly compared to metabolic rates of the upstream, degraded river (measured at station D with the 1-station open channel method). Results obtained with the 1-station and the 2-station method often agree well (e.g., Bernot et al., 2010; Beaulieu et al., 2013). Thus, we quantitatively supported our previous qualitative findings by estimating metabolism with the 2-station method for the combined restored reach R1+R2. We would like to stress that the good agreement of results of the 1-station and the 2-station method in restored reaches, i.e. that both clearly suggested metabolism increases due to restoration, may be an important finding for agency efforts to monitor restoration outcome, because the 1-station method may be more practical for routine measurements, while the 2-station technique is often considered a research method that is too complex for such purposes. Therefore, we believe that it is important to show and compare results from both techniques in our study.

COMMENT
Time series analyses could be a more powerful way quantify differing/strengths of controls on metabolism in R vs D reaches during the 50-day deployments. A more sophisticated analysis would enhance the contribution of the paper beyond means/ranges of GPP, ER, NEP. See Roley et al 2014 Freshwater Science for an example of this and a citation of general interest. If not used here, this is at least worth a mention for future research avenues.

RESPONSE
We changed our statistical approach and now use an autoregressive approach to address the autocorrelation issue of time-series data (see response to reviewer 1 for a more detailed description). However, this new approach only resulted in minor changes in results.

COMMENT
The importance of autochthonous production for ecosystem functioning" (19-20) is a very context-dependent statement that should be expressed with caution. What does an increase in GPP mean for ecosystem health? We see this response in the R2 reach (but D was 2nd highest in macrophytes, not R1?), and higher GPP is sometimes used as a sign of ecosystem degradation: excess primary production in response to nutrient loading. Units of N are in mg, so there may be environmental issues requiring mitigation beyond physical restoration by widening river channels. Without "pre" data for R reaches, one could argue that the restoration project provided more light + warmer temperatures needed for enhancement of macrophytes beyond "natural" conditions.

RESPONSE
We express increased abundance of macrophytes as positive effect of restoration (see Lorenz et al. 2012 J. Appl. Ecol.) rather than using it as indicator for high nutrient load and eutrophication. Restoration aimed to establish near-natural hydromorphology and biota. Thus, increased occurrence of macrophytes is positive effect in terms of near-natural state of the investigated river. See also response to comment of reviewer 1: We agree and discussed this issue in more detail in the revised paper. In the present case, there are no point or diffuse sources that lead to eutrophication effects in this river section. Thus, metabolic responses should be solely due to river restoration, i.e. wider channels (-> more light availability), shallower channels (-> better habitat for macrophytes), and lower current velocities (->less hydraulic stress) as also discussed in lines 70 of the original manuscript. In R1 other autotrophs (e.g., periphyton) which have not been subject to mapping may have stimulated metabolism.

COMMENT
L2 - No comma needed after "Both"

RESPONSE
We agree and corrected this in the revised paper.

COMMENT
L - 49-50 – Should include a "but see: : :" citation here to acknowledge that there have indeed been several previous studies even if this issue is not well-studied or settled.

RESPONSE
We have added this to lines 53ff of the original manuscript.

COMMENT
L - 53 – Delete "only"

RESPONSE
We corrected this in the revised paper.

COMMENT
L - 109 – It would be useful to give time frame of/since restoration here, not only in supplement.

RESPONSE
We agree and added this information in the next paragraph of the revised paper. Please see also response to reviewer 3.

COMMENT
L - 130 – What about water chemistry? Any differences among reaches? If restoration is indeed altering nutrient retention or removal, that should be reflected in downstream concentrations.

RESPONSE
It may well be possible that changes nutrient concentrations are measurable along the >2 km of restored river. However, we did not measure nutrient concentrations along the studied stream section, and therefore cannot comment on that.

COMMENT
L - 183-4 – Vague RE: which methods as written. Why not restate and give equations for this, k, and base metabolism calculation to allow readers to better assess the methods within the manuscript itself?

RESPONSE
We agree with the reviewer that it is interesting to have the mentioned equations and added them to the manuscript.

COMMENT
L - 225 – Good to see that the authors will update their statistical analyses in a revised manuscript.

RESPONSE
The detailed results from these analyses are in our response to reviewer 1.

COMMENT
L - 227 – Data from flood events are one of the most exciting things we can learn about from longer time series. I urge the authors not to exclude them from their analyses.

RESPONSE
We agree that floods and storms are an important part of environmental variability. However, GPP was not detectable during storm events, and we cannot be sure whether GPP was really zero or very low, or whether high flows may have prevented the detection of GPP. As reviewer 1 also stated, this often occurs with high stream discharge events. Thus, we show the data and discuss them in terms of environmental variability, but believe that it is more adequate not to analyze flood data statistically in terms of restoration effect.

COMMENT
L - 273 – "returned"

RESPONSE
We agree and corrected this in the revised paper.

COMMENT
L - 347 – Give some numbers to support this comparison in the main text. Fine to keep the citations/table in supplement.

RESPONSE
We agree and added more detailed information in the discussion.

COMMENT
L - 369 – "near-natural" is a very vague description and does not seem to be supported with data.

RESPONSE
We removed the term „near-natural" conditions ("Thus, the restoration of short river reaches may have positive effects…").

COMMENT
L - 380 – "as a functional indicator".

RESPONSE
We agree and corrected this in the revised paper.

COMMENT
Table 2 – Check units. Superscripts for m didn't appear in my version of the text.

RESPONSE
In our version of the manuscript superscripts are correctly represented. We confirmed this in the final version.

COMMENT
Fig 2 – Needed? There are many figures, and I didn't feel that this was needed for main text. Supplemental figures are nice.

RESPONSE
We believe that these figures are important to understand our rationale. Moreover, as Biogeosciences is an online journal without figure limits, we thought that this would not be a problem.

COMMENT
App S5 – Possible to include Q for context? See also Genzoli & Hall 2016 FWS, Davis et al 2012 RRA, Dodds et al. 2013 FWS, Hall et al 2015 L&O.

RESPONSE
Yes, we included Q in spreadsheet (a) "river characteristics" of Appendix 5.

**Detailed response to reviewer 3**

COMMENT
This manuscript examines the response of ecosystem metabolism to river restoration by comparing ecosystem metabolism among three river reaches of a mid-size river: a degraded (unrestored) reach (D), a moderately restored (R1), and a substantially restored reach (R2). The use of ecosystem metabolism to determine the effects of river restoration is fairly novel. In that sense, the manuscript represents a relevant contribution to the challenge of incorporating measures of ecosystem functioning to river monitoring, and to river restoration in particular. The manuscript is well structured and written. In general, the results are clearly and transparently exposed, limitations indicated, and details nicely presented in the appendixes. My main concern with this manuscript relies on the fact the sampling design of this case study may not be the most appropriate for correctly answering the important question of whether river restoration caused a significant change in ecosystem metabolism. Ecosystem metabolism was measured in only one river, only after river restoration, and only during a certain period of the year (summer). Ideally, such a question should have been addressed by measuring in several rivers, before and after restoration (BACI design), and considering several periods of the year. None of these criteria is fulfilled, and therefore, the strength of the results and its potential extrapolation to more general responses are limited. This should be at least more explicitly acknowledged in a revised version of the manuscript.

RESPONSE
The comparison of restored with upstream degraded "control" sections (space-for-time substitution) is commonly used to quantify restoration effects as data on pre-restoration are rare (e.g., Hering et al. 2015 J. Appl. Ecol., Jähnig et al. 2011 J. Appl. Ecol.). Time of measurements is in accordance with the WFD compliant sampling period for structural measures such as macroinvertebrates and macrophytes. Consequently, our results can be linked to these measures (see lines 315-318).
The measurements are extensive and time consuming and it was not the aim of our paper to replicate measurements in several rivers – this would have required a much simpler method, thus gaining limited insights into the underlying mechanisms. We would like to stress that our manuscript is the first measuring metabolic changes of medium-sized rivers following restoration. The lack of available data - especially for mid-sized and larger rivers - is discussed in lines 351-356 of the original manuscript.

COMMENT
I agree with the comments that have arisen in the open discussion regarding the use of the 1-station method. As indicated in those comments and responses, the authors should incorporate the 2-station method for the restored reaches (R1 + R2) to make their statements more robust. These limitations in the metabolism estimations together with the issue in the general sampling design (previous paragraph) and the statistically significant but relatively minor changes in metabolic fluxes in restored relative to degraded reaches, makes the conclusions of a clear effect of the restoration on ecosystem metabolism not as clear as pointed out by the authors.

RESPONSE
Please see our response to reviewer 1. We do not consider significant GPP increases from ≈5 to ≈7 g DO $m^{-2}$ $d^{-1}$ and significant NEP increases from ≈-2 to ≈0 g DO $m^{-2}$ $d^{-1}$ as minor changes from an ecosystem perspective.

COMMENT
I also think that the last part of the discussion could be greatly improved by making more explicit recommendations and by being more convincing about the advantages of incorporating metabolism and other functional measures to river monitoring.

RESPONSE
We agree with the reviewer that advantages of functional indicators for river monitoring are manifold and should be mentioned in studies, which are using functional measures in applied river research. However, we feel that this topic - and ecosystem metabolism in particular - has been widely discussed within the literature (see also references presented in line 382 of the original manuscript) and that a summary of the benefits of metabolism as a functional indicator is more appropriate as presented in lines 380 - 387 of the original manuscript. We think that a more extensive description would make the manuscript lengthier without giving new information.

COMMENT
L - 17-18: Unclear sentence. Rephrase.

RESPONSE
We changed the sentence as follows: "Restoration increased autotrophic processes, as indicated by higher GPP:ER rates measured at restored reaches".

COMMENT
L - 23-24: Any hints that this is occurring at the study site?

RESPONSE
"High rates of metabolism and the occurrence of dense macrophyte stands may increase the assimilation of dissolved nutrients and the sedimentation of particulate nutrients, thereby positively affecting water quality." – This is our interpretation of possible consequences of the observed high metabolism rates, although we did not measure nutrients permanently. To highlight the speculative nature of this sentence we use the word "may".

COMMENT
L - 61-64: "natural changes" and "land-use change" are confusingly used in this sentence.

RESPONSE
We think the difference is clearly described by giving examples in the original manuscript (see lines 62 and 64). Natural changes refer to floods and droughts (e.g., Uehlinger, 2000) and land-use change refers to differences between pristine and agricultural streams (e.g., Gücker et al., 2009; Silva-Junior et al., 2014).

COMMENT
L - 79: Any reference?

RESPONSE
"Moreover, the reconnection of rivers with their floodplains by creating shallower river profiles and removing bank fixations may enhance inundation frequency, and hence resource transfers from land to water" - This is our interpretation of possible consequences of the hydromorphological restoration (reconnection of river and floodplain by creating shallower river profiles and removing bank fixations) although we are not aware of a reference. To highlight the speculative nature of this sentence we use the word "may".

COMMENT
L - 89: Do you mean "contiguous" instead of "continuous"?

RESPONSE
We agree and corrected this in the revised paper.

COMMENT
L - 91: Here I miss some predictions regarding the expected differences between R1 and R2. It seems important to justify the examination of two levels of restoration.

RESPONSE
We expected (i) hydromorphological river characteristics, i.e. habitat composition and hydrodynamics, to change following restoration, with the magnitude of change depending on restoration effort (e.g. width and diversity of the river channel, and abundance of primary producers, as well as sizes and locations of transient storage zones in the two restored river reaches compared to the degraded reach).

COMMENT
L - 110-115: I suggest including here information on when the restoration was done. It seems important to know how much time has passed from restoration to measurements.

RESPONSE
We agree and added this information in the revised paper. See also response to reviewer 2.

COMMENT
L - 138: Unclear at which flow conditions these measures were done.

RESPONSE
We agree and added this information in the revised paper.

COMMENT
L - 195: It seems odds that some measures were done in 2013 and others in 2014. How may this have influenced your results?

RESPONSE
The purpose of our study was to quantify structural changes following restoration at the reach scale and examine the related effects on metabolism. Considering that there were no major meteorological differences between these years, that structural differences between D, R1 and R2 (effort) were very similar between years, and that measurements were performed in the same season at similar discharge conditions, we believe that interannual variability is a minor confounding factor in our study.

**Detailed response to Robert Pennington**

We agree with you that the estimation of the reaeration rate is probably the most critical step in the estimation of open-channel metabolism, and deserves further discussion. In order to present a short and concise paper, we tried not to overburden our paper with methodological details, but we are happy to discuss your suggestion (see point-by-point response below) and to add this information to our paper.

COMMENT:
One point I would like to see more complete discussion of is how the aeration rates were calculated. It would be nice to see a plot of $dDO/dt = ER + K(DO\_deficit)$ for a few nights that were considered significant.

RESPONSE:
The nighttime regression procedure is a rather standard technique, widely used in open-channel metabolism studies (e.g., Young & Huryn, Ecol. Appl. 9: 1359-1376). This technique is particularly suitable for rivers with considerable GPP that causes considerable daytime DO increases, and stable nighttime DO plateau. It is suitable for streams with $K_{oxy} < 0.5$ m h-1, such as the investigated river (see Demars et al., Limnol. Oceanogr.: Methods 13, 356–374). Below we show two plots (the better fit with $p < 0.01$ and $R^2 = 0.69$ and the worse one with $p < 0.05$ and $R^2 = 0.33$) from the first two sampling weeks at station R2 to give an impression about the variability in the encountered nighttime patterns. The regression slope corresponds to the $K_{oxy}$ (in 1/s) in these representations. In our study, we only considered significant nighttime regressions ($p < 0.05$).

[Figure]

[Figure]

COMMENT:
Adding confusion to the aeration rate discussion is the reported units of appendix S2 in g O2/(m3*s). Is this a typo? Why not use the same units as the text in 1/day (line 215)?

RESPONSE:
Yes, this is a typo. The y-axis units for plots in S2 should be 1/day. Thank you!

COMMENT:
Also, given the relatively low aeration rate and high productivity, why not use a parameter fitting approach to model metabolism and aeration rate? It seems a more robust approach than the nighttime regression method.

RESPONSE:
Inverse modeling approaches, such as BaMM (Holtgrieve et al., Limnol. Oceanogr., 55: 1047–1063) and BASE (Grace et al., Limnol. Oceanogr.: Methods 13: 103–114) require PAR irradiance input data for the simultaneous estimation of reaeration and metabolism. Unfortunately, we do not have adequate PAR data available for the studied river. Whether inverse modeling yields more reliable reaeration estimates than the nighttime regression method (and under which circumstances?) has not been tested in the literature to our knowledge. In our opinion, the best approach would be to obtain reaeration rates from different methods (gas tracer experiments, nighttime regression, inverse modeling) in future studies. However, we believe that our reaeration estimates (ranging from about 6/d to 16/d across 3 sampling stations and 50 sampling days) are realistic estimates for the investigated river.

COMMENT:
Your study finds very high GPP and ER estimates compared to others. This could be a direct result of an overestimate of the aeration rate. I find that a more robust method, or convincing discussion of the aeration rate, is necessary to support these findings. If an entirely new analysis applying a parameter fitting model is perhaps infeasible, empirical values from hydrodynamics and morphology would be helpful.

RESPONSE:
To our knowledge, the nighttime regression technique has not been shown to be less robust than other methods in the literature. We see the available methods rather as complementary techniques. For example, gas tracer approaches are considered a direct and reliable method to measure reaeration, but represent only a temporal snapshot, and can yield unreliable results if conservative tracer dilution is not adequately quantified. The nighttime regression method estimates reaeration over a longer time scale and has been found to work well in less turbulent, productive streams, but is often not very useful in turbulent, shaded headwater streams. Inverse modeling may be the most integrative approach, but measured diel DO and temperature data can sometimes be uninformative and prior information (such as field tracer studies) is needed (see Holtgrieve et al., Limnol. Oceanogr.: Methods 14: 110–113). However, if not established for a specific sampling site, empirical equations may be the most uncertain option to estimate reaeration, as results of different empirical equations vary widely (at least 40-125% error in estimates, see Demars et al., Limnol. Oceanogr.: Methods 13, 356–374). Considering two contrasting hydromorphological parameter sets from our study (20 June 2014, low discharge, 9.1 m3/s; and 12 July 2014, high discharge 32.7 m3/s), reaeration rates estimated by the two most commonly used empirical equations indeed varied considerably, but yielded consistently higher reaeration rates than our nighttime regressions for these days (Table 1).

**Table 1:** Comparison of reaeration coefficients estimated in this study with those estimates from empirical models for the same reaches.

| Reach | Owens 1974 (surface renewal model) $K_{oxy}^{20}$ (/d) | Tsivoglou & Neil 1976 (energy dissipation model) $K_{oxy}^{20}$ (/d) | Nightime regression $K_{oxy}^{20}$ (/d) |
|---|---|---|---|
| Low Q | | | |
| D | 29 | 42 | 11 |
| R1 | 51 | 36 | 9 |
| R2 | 17 | 19 | 7 |
| High Q | | | |
| D | 48 | 127 | 11 |
| R1 | 30 | 73 | 9 |
| R2 | 15 | 45 | 9 |

Therefore, we do not think that overestimations of reaeration rates explain the high metabolic rates in our study. As discussed in the paper, we measured river metabolism during the peak of macrophyte growth in summer. This may explain the high rates of GPP and R in our study, and consequently, lower rates may be expected in other seasons.

COMMENT:
Also, would it be possible to apply the two-station method if you combined reaches? My rough calculations indicate a reach length of about 3000 meters would be appropriate from your hydraulics and aeration rate.

RESPONSE:
According to Demars et al. (Limnol. Oceanogr.: Methods 13, 356–374), the 2-station method is applicable to reach lengths 0.4 v/k to 1.0 v/k. In our reach system, this range corresponds to:

| | |
|---|---|
| D | 3283 - 8280 m |
| R1 | 2765 - 6912 m |
| R2 | 1624 - 4061 m |
| ------------------------------------------------------------ | |
| R1+R2 | 2199 - 5497 m |
| all 4 reaches | 2482 - 6205 m |

Thus, we cannot evaluate separate reaches in our study, but are at the lower end of this range (slightly above 0.4 v/k) for all 4 reaches combined and for R1+R2, and can therefore estimate 2-station metabolism for these combinations. Combining all reaches is of limited use for the aim of our paper to evaluate restoration effects, but combining restored reaches R1 and R2 indeed provides useful information. In response to your comment, we evaluated 2-station metabolism for R1+R2. Below please find the results of the application of the 2-station method for the first sampling day (June 20, 2014). Rates of GPP were only slightly higher for the combined reaches R1+R2 using the 2-station method than for the longer reach evaluated by the 1-station method at station R2 (that included a longer unrestored upstream section), i.e. 11.4 vs. 8.8 g DO m-2 d-1. Rates of R, were only slightly lower with the 2-station method, i.e. 9.6 vs. 11.2 g DO m-2 d-1. These results further support our hypothesis of increased river metabolism due to restoration. In the revised manuscript, we used the 2-station method systematically for the combined reaches R1+R2 for all sampling days.

| Ecosystem Respiration | 9.59 | g O2/m2 d |
|---|---|---|
| Gross Primary Production | 11.36 | g O2/m2 d |
| Net Ecosystem Production | 1.76 | g O2/m2 d |
| P:R | 1.18 | |

(values on y axis in g DO m-2 d-1 /1000)

[revised manuscript text omitted]